# Innate and adaptive signals enhance differentiation and expansion of dual-antibody autoreactive B cells in lupus

Allison Sang[1], Thomas Danhorn [2], Jacob N. Peterson[1], Andrew L. Rankin[3,4], Brian P. O'Connor[1,2,5,6], Sonia M. Leach[2,5], Raul M. Torres[1,5] & Roberta Pelanda[1,5]

Autoreactive B cells have a major function in autoimmunity. A small subset of B cells expressing two distinct B-cell-antigen-receptors ($B_{2R}$ cells) is elevated in many patients with systematic lupus erythematosus (SLE) and in the MRL(/lpr) mouse model of lupus, and is often autoreactive. Here we show, using RNAseq and in vitro and in vivo analyses, signals that are required for promoting $B_{2R}$ cell numbers and effector function in autoimmune mice. Compared with conventional B cells, $B_{2R}$ cells are more responsive to Toll-like receptor 7/9 and type I/II interferon treatment, display higher levels of MHCII and co-receptors, and depend on IL-21 for their homeostasis; moreover they expand better upon T cell-dependent antigen stimulation, and mount a more robust memory response, which are characteristics essential for enhanced (auto)immune responses. Our findings thus provide insights on the stimuli for the expansion of an autoreactive B cell subset that may contribute to the etiology of SLE.

[1] Department of Immunology and Microbiology, University of Colorado School of Medicine, Aurora, CO 80045, USA. [2] Center for Genes, Environment and Health, National Jewish Health, Denver, CO 80206, USA. [3] Inflammation and Immunology, Pfizer Research, Cambridge, MA 02140, USA. [4] Immuno-Oncology Discovery, FivePrime Therapeutics, South San Francisco, CA 94080, USA. [5] Department of Biomedical Research, National Jewish Health, Denver, CO 80206, USA. [6] Department of Pediatrics, National Jewish Health, Denver, CO 80206, USA. Correspondence and requests for materials should be addressed to R.P. (email: Roberta.Pelanda@ucdenver.edu)

A hallmark of systemic lupus erythematosus (SLE) is a breach in B cell tolerance that results in the production of autoantibodies. Autoantibody-producing B cells play additional pathogenic roles in the progression of SLE by secreting inflammatory cytokines and directly activating pathogenic T cells[1–3]. Autoreactive B cells originate from the bone marrow via expression of newly rearranged germline immunoglobulin (Ig) genes but also arise from peripheral lymphoid tissue subsequent to the somatic hypermutation of Ig genes. During the development of B cells in the bone marrow, random Ig gene rearrangements generate a large number of autoreactive B cells[4,5]. However, at least half of these cells are immediately eliminated by receptor editing (i.e., secondary Ig gene recombination) or clonal deletion[6,7,8]. Receptor editing most often results in the elimination of the autoreactive specificity displayed by the B cell antigen receptor (BCR) and the expression of a new nonautoreactive BCR. However, receptor editing can also lead to the generation of dual-reactive ($B_{2R}$) B cells: B cells co-expressing two different heavy (H) or light (L, κ or λ) Ig chains[9] and, thus, two different BCRs. These BCRs are comprised of the initial autoreactive antigen receptor and a new nonautoreactive receptor most often based on the association of the original H chain with a new L chain. $B_{2R}$ cells have a demonstrated ability to bypass tolerance checkpoints and escape to the periphery where they are more or less regulated by mechanisms of peripheral tolerance, depending on the genetic background and the level of autoreactivity of the B cell clone[10–16]. Elegant recent studies indicate that many of the B cells that participate in disease flares in lupus patients are naive B cells with germline Igs[17] and they are therefore generated from de novo B cell development, as opposed to arising from a germinal center (GC) reaction. Thus, the establishment of central (bone marrow) B cell tolerance is of vital importance for the prevention and/or control of autoimmunity in humans.

$B_{2R}$ cells are observed in both healthy mice and humans at a frequency of less than 3% of all B cells[14,18–22]. Our previous findings have shown that the fate of $B_{2R}$ cells can be different in an autoimmune background; here these cells expand with disease and show high level of activation[23]. To study $B_{2R}$ cells we have employed the use of congenic mice bearing a gene targeted human Ig Cκ allele[24] on a healthy (CB17) or an autoimmune (MRL) genetic background[23]. These $Igk^{m/h}$ mice allow for the detection of B cells co-expressing two different κ chains (dual-κ) within a wild-type Ig repertoire. We have previously found that dual-κ B cells accumulate in both MRL and MRL/lpr mice with age, while this is not observed in the non-autoimmune CB17 mice[23]. Furthermore, dual-κ B cell enrichment in MRL(/lpr) mice correlates with disease progression and with the appearance of autoantibodies. These $B_{2R}$ cells are particularly enriched in the effector plasmablast and IgG$^+$ memory B cell compartments of MRL(/lpr) mice, where they represent up to 50% of cells in older mice. Overall, dual-κ B cells are more autoreactive, express higher levels of activation markers, and secrete larger amounts of autoantibodies than single-κ B cells[23]. Interestingly, dual-κ B cells do not display these characteristics in NZB/NZW lupus mice[25] and, thus, their association with lupus-like disease is likely dependent on genetic polymorphisms. Similar to what has been described in lupus mice, a recent study found that $B_{2R}$ cells co-expressing κ and λ, are expanded in a subset (about 40%) of SLE patients[26]. This suggests $B_{2R}$ cells are a significant player of SLE pathogenesis and that, gaining a better understanding of their biology, is important.

In this study we exploited our dual-κ autoimmune mouse model (MRL/lpr-$Igk^{m/h}$) to identify molecular pathways that can drive the accumulation and activation of $B_{2R}$ cells in murine lupus. We show that a significant difference exists between single and dual-κ B cells both functionally and at the transcriptome level. Overall, our data indicate that both T cell-mediated signals and innate stimuli, together and independently, play a role in the enrichment, activation, and differentiation of dual-κ B cells during lupus-like disease in MRL/lpr mice.

## Results

**MRL/lpr dual-κ B cells proliferate more than single-κ cells**. We have previously shown that dual-κ B cells from MRL and MRL/lpr lupus mice are enriched in the plasmablast cell compartment[23], which is a stage at which cells are actively proliferating. To evaluate whether this plasmablast enrichment is due to an increased proliferative ability of dual-κ relative to single-κ B cells, we analyzed the expression of the proliferation marker Ki67 and the frequency of EdU incorporation, as readouts for active cell division. The frequency of Ki67$^+$ dual-κ CD138$^-$ B cells was 2–3-fold greater than that of single-κ cells in the spleen of naive MRL/lpr-$Igk^{m/h}$ mice, while it was similar in the CD138$^+$ plasmablast compartment (Fig. 1a, b and Supplementary Fig. 1a, b). Furthermore, 24 h after the injection of EdU, the frequency of EdU$^+$ dual-κ splenic B cells was 3-fold higher than that of single-κ B cells (Fig. 1c, d and Supplementary Fig. 1a–c), thus indicating enhanced active proliferation of dual-κ B cells in vivo.

Previous studies have shown that c-Myc is rapidly upregulated by mature B cells following BCR engagement[27] and that this event leads to cell proliferation[28]. In line with the higher proliferation rates described above, a larger frequency of dual-κ B cells than single-κ B cells expressed c-Myc protein (Fig. 1e, f and Supplementary Fig. 1a–d).

Collectively, these results indicate that under homeostatic conditions, dual-κ B cells of MRL/lpr mice proliferate at a higher rate than single-κ B cells and possibly as a consequence of antigen engagement and c-Myc expression.

**RNAseq and pathway analyses of single and dual-κ B cells**. To identify genes and pathways involved in the enhanced activation and proliferation of dual-κ B cells in murine lupus, we performed RNAseq analyses of single and dual-κ B cells from MRL/lpr-$Igk^{m/h}$ mice. Because dual-κ B cells are biased toward the marginal zone (MZ) subset[23], in order to prevent an artificial skewing we analyzed single and dual-κ follicular (FO) and MZ splenic B cell populations, sorted as described in Supplementary Fig. 2a. RNAseq analyses identified 1938 and 446 genes (false discovery rate, FDR ≤ 0.05) in FO and MZ B cells, respectively, that were differentially expressed in dual-κ B cells relative to single-κ B cells (Fig. 2a). The overexpression of Ig Vκ genes and not of CD19 and CD79A genes in dual-κ cell samples, verified the accuracy of the cells used for the RNAseq data (Fig. 2b). In addition, Jκ1 RNA amount was decreased in dual-κ B cells while Jκ5 was increased (Fig. 2c), a result that is consistent with higher frequency of receptor editing[29]. We also analyzed the abundance of CD138 and IgG1, IgG2b, and IgG2c transcripts in the FO B cell samples to estimate the level of contamination by plasmablasts and IgG-switched cells. One of the samples had a larger amount of these transcripts, but the contamination was generally similar in the dual-κ and single-κ samples (Supplementary Fig. 2b).

We next used ingenuity pathway analysis (IPA) to identify pathways that are specifically guiding the biology of dual-κ B cells. The differentially enriched pathways with particular immunological significance include BAFF and APRIL-mediated signaling, Toll-like receptor signaling, IL-6 signaling, MAPK signaling, and antigen presentation, for both FO and MZ B cells (Fig. 2d). The top 30 pathways detected by IPA in FO B cells (-log (p-value) > 6.05) included pathways involved in antigen and cytokine receptor signaling, such as B Cell Receptor, TNFR1/2, mTOR, PI3K/AKT, CD27, NFAT, JAK/Stat, and JAK1, JAK2,

and TYK2 signaling in interferon. On the other hand, the pathways with B-cell relevance detected in MZ B cells among the top 30 (-log($p$-value) > 4.53) were role of pattern recognition receptors in recognition of bacteria and viruses, complement system, IL-6 signaling, and toll-like receptor signaling. Thus, while there was overlap in the RNAseq data from FO and MZ B cells, there were also some distinctions.

Results from these gene expression analyses suggest dual-κ B cells from MRL/*lpr* mice possess a differential ability to respond to T cell-dependent and independent signals relative to single-κ B cells.

**Dual-κ cells show higher responses to innate stimuli.** The TLR and IFN I and II pathways are of particular relevance for the onset and progression of lupus disease[2,30,31]. Based on the RNAseq analysis, dual-κ FO B cells displayed differential expression of 21 out of 76 genes and 14 out of 36 genes in the TLR signaling and the interferon signaling pathways, respectively (Fig. 3a). To validate these results, we investigated in vitro responses of single and dual-κ B cells to these innate stimuli.

Untouched (CD43⁻) B cells isolated from the spleen of MRL/*lpr*-*Igk*^m/h^ mice were cultured in the presence or absence of the

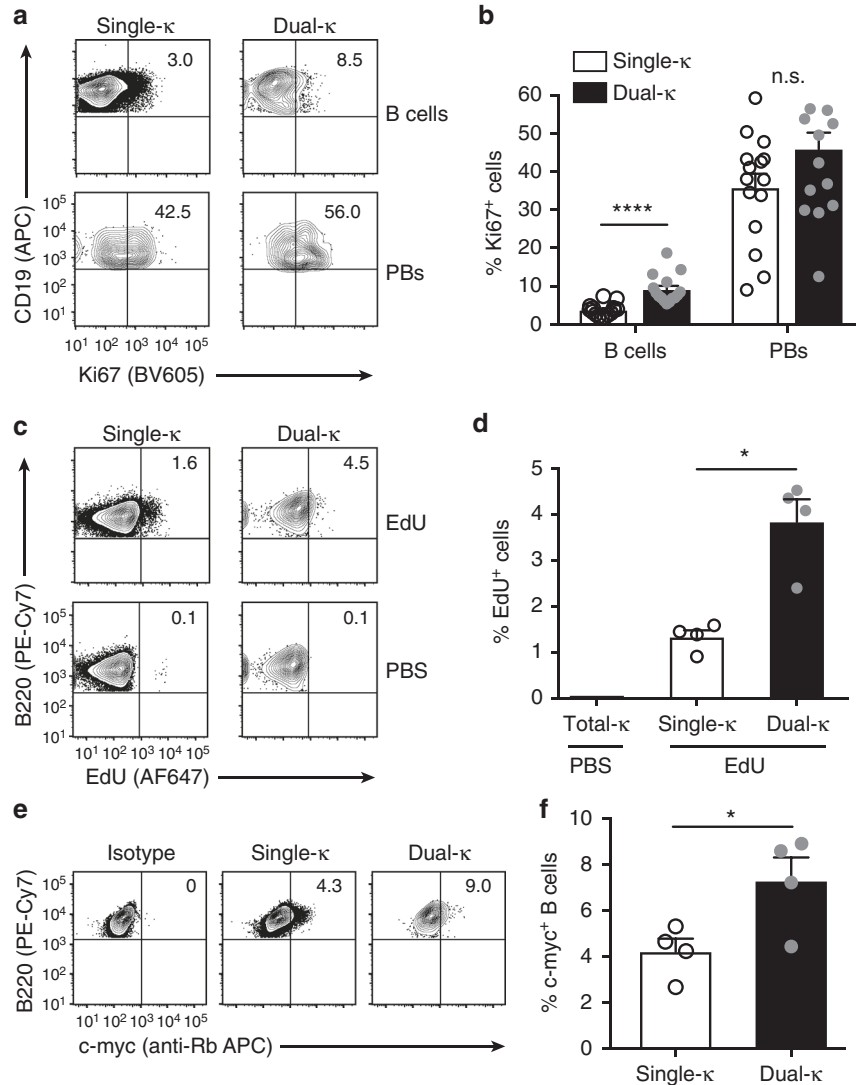

**Fig. 1** Splenic dual-κ B cells display higher proliferation rates than single-κ B cells. **a** Representative flow cytometric contour plots showing frequency of Ki67+ single and dual-κ cells within B220+CD19+ CD138⁻ B cells (top) and CD138+CD44^high^ plasmablasts (PBs, bottom), gated as shown in Fig. S1a, b. **b** Mean frequency of Ki67+ cells within single (white bar) or dual-κ (black bar) B cells and PBs described in **a**. Data are combined from three independent experiments using 9–16-wk-old MRL/*lpr*-*Igk*^m/h^ mice (N = 15 total). **c** Representative contour plots showing the frequency of EdU+ single and dual-κ cells within CD3⁻B220+ B cells (gated as in Supplementary Fig. 1a–c) 24 h after either EdU (top row) or PBS (bottom row) injection in MRL/*lpr*-*Igk*^m/h^ mice. **d** Mean frequency of EdU incorporation by splenic single-κ (white bar) or dual-κ (black bar) B cells (gated as described in **c**), after EdU injection in 12–14-wk-old MRL/*lpr*-*Igk*^m/h^ mice. Data from one untreated MRL/*lpr*-*Igk*^m/h^ mouse indicate EdU background staining in total Igκ+ B cells. Representative data from one out of three experiments is shown with N = 4 in each experiment. **e** Representative flow cytometric analysis to measure the frequency of c-Myc-positive cells within splenic CD3⁻B220+ single-κ (middle panel) or dual-κ B cells (far right panel), gated as shown in Supplementary Fig. 1a–d. Far left panel shows Igκ+ B cells stained with an isotype control. **f** Mean frequency of c-Myc+ cells within single or dual-κ B cell subsets described in **e** in 13-wk-old MRL/*lpr*-*Igk*^m/h^ mice. N = 4 from one experiment. *P < 0.05, ****P < 0.0001; n.s. not significant. Significance was assessed by Student's *t*- or Mann–Whitney tests. In all bar graphs the standard error bars represent SEM and each symbol an individual mouse

TLR ligands CpG (for TLR9), R848 (for TLR7), and LPS (for TLR4), or with IFN-α (type I) or IFN-γ (type II) to measure their contribution to the relative frequency and activation of single and dual-κ B cells. TLR7 and TLR9 stimulation significantly expanded dual-κ B cells over single-κ B cells, while TLR4 stimulation (LPS) had no effect (Fig. 3b and Supplementary Fig. 3a, b). Despite this difference, single and dual-κ B cells similarly increased CD69 expression (Fig. 3c), suggesting that

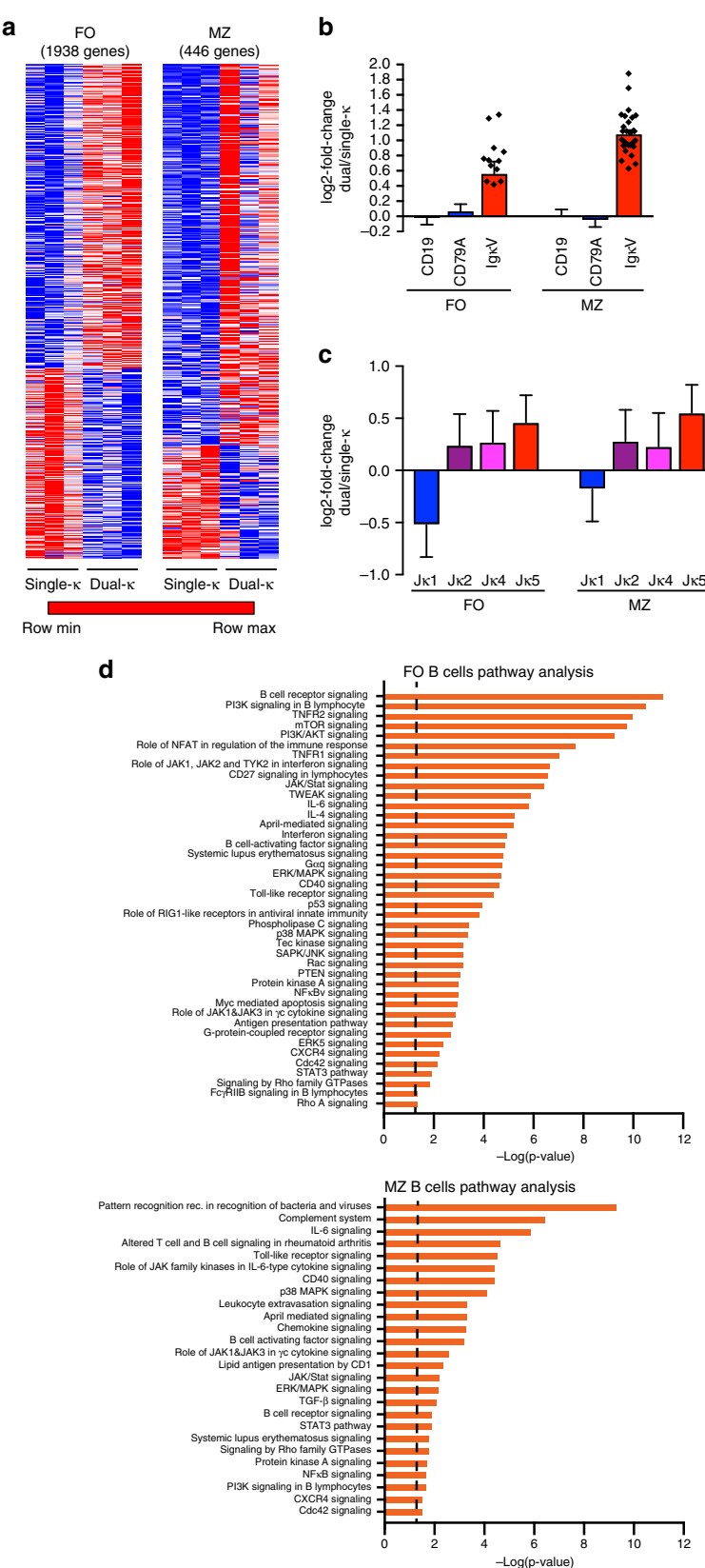

TLR7 and TLR9 signaling preferentially promotes the expansion, but not the activation of dual-κ B cells relative to single-κ B cells. In contrast, culturing B cells with either IFN-α or IFN-γ induced significantly higher expression of CD69 (Fig. 3d and Supplementary Fig. 3a) and CD86 (Supplementary Fig. 3c) on dual-κ relative to single-κ B cells, although neither IFN altered the relative frequency of dual-κ B cells (Fig. 3e). This indicates that IFNs I and II may preferentially promote the activation and not the expansion of dual-κ B cells. The pathway analysis also highlighted a differential activation of BAFF and APRIL signaling, but incubation with these two B cell survival factors did not promote any preferential expansion of dual-κ B cells (Supplementary Fig. 3d, e).

We have previously shown that the splenic dual-κ B cell population of MRL/*lpr* mice holds a larger frequency of activated B cells than the single-κ population[23]. We wondered if MRL/*lpr* dual-κ B cells display an increased activation starting from their origin, and whether their increased activation accounts for their differential responses to TLR agonists and IFNs described above. To address the first issue we analyzed the relative expression of the CD80 and CD86 activation markers on immature B cells of mice at 8 and 14 weeks of age. The expression of CD80 and CD86 on newly generated dual-κ B cells was larger than that on single-κ B cells, while there was little or no differential expression of CD138 (Supplementary Fig. 4a, b). This suggests that from their onset, dual-κ B cells are inherently more activated. To directly test whether the increased activation of dual-κ B cells skewed the in vitro responses to innate stimuli, we analyzed the responses of untouched naive B cells, cells that were magnetically sorted from the spleen as CD43⁻CD80⁻CD86⁻ (Supplementary Fig. 4c). Even when comparing the naive B cell population, dual-κ B cells displayed enhanced responses (increased frequency and/or activation) to CpG, R848, and IFN-α relative to single-κ B cells (Fig. 3f, g).

Together, these data indicate that acute innate stimuli such as TLR and interferon receptor signaling play a role in the preferential enrichment and/or the increased activation of dual-κ B cells in MRL/*lpr* mice.

**Dual-κ B cells display higher amounts of co-receptors**. IPA indicated an enrichment of the antigen presentation pathway in dual-κ FO B cells (Fig. 2d). To assess the role T cells may play in the activation of dual-κ B cells, we analyzed the expression of several co-receptors that B cells use for cognate engagement with T cells.

In agreement with the RNAseq analysis, dual-κ B cells expressed significantly higher amounts of surface MHCII proteins (Fig. 4a, b). This difference was more pronounced in the FO than the MZ B cell compartment, whereas no difference was observed at the plasmablast stage (Fig. 4c). The RNAseq analysis also reported increased *IL-21R* RNA transcripts in dual-κ relative to

single-κ FO B cells, but not in MZ B cells (Fig. 4d). This is of interest since IL-21 is a cytokine that is secreted by T FO (and extra-FO) helper ($T_{FH}$) cells to play a critical role in the formation of GC B cells, Ig class switched B cells, and plasma cells[32,33]. Indeed, dual-κ B cells expressing IL-21R were at higher frequency (Fig. 4e, f) and expressed higher levels (Fig. 4g) of this receptor when compared to single-κ cells.

Dual-κ B cells also showed significant upregulation of the T cell co-stimulatory molecules CD80, CD86, and PD-L1 relative to single-κ cells (Fig. 4h, i and Supplementary Fig. 5a), while no difference was seen in the expression of the B cell co-receptor CD19 (Fig. 4i). In contrast, expression of the co-stimulatory molecule CD40, which is crucial for receiving T cell help, was significantly lower when measured on total dual-κ B cells (Fig. 4i), but this appeared to be due to an expansion of a CD40^low cell subset (Fig. 4h) that we identified to consist of plasmablasts (Supplementary Fig. 5b). Indeed, CD40 levels on CD40⁺ B cells were slightly significantly higher on dual-κ than single-κ cells (Fig. 4h, i). This difference appeared to be functional, since stimulation of B cells with anti-CD40 antibodies resulted in higher levels of CD69 (but not CD86) on dual-κ B cells (Supplementary Fig. 5c). Cognate interaction of B and T cells leads to the formation of GCs. Relatedly, the frequency of PNA^high CD38^low GC B cells within the dual-κ population was more than double than that observed in the single-κ cell subset (Fig. 4j, k). IgD levels, which can be downregulated on antigen-exposed B cells, where 10% decreased on FO dual-κ relative to single-κ cells, while IgM was slightly upregulated, but these differences were not significant (Supplementary Fig. 5d, e).

Overall, these data indicate that dual-κ B cells upregulate many if not most of the known adaptive co-stimulatory receptors and, thus, are poised to receive increased T cell help.

**Contribution of IL-21R to the differentiation of dual-κ B cells**. The development of lupus disease is significantly inhibited in MRL/*lpr* mice deficient for IL-21R[34]. IL-21, a cytokine secreted by CD4 $T_{FH}$ cells, promotes the generation of plasma cells and the development of IgG class switched antibodies and autoantibodies[32,33]. Given the increased expression of IL-21R by dual-κ B cells (Fig. 4e–g), we questioned whether IL-21 contributes to the positive selection of these cells into effector populations. To study this, we crossed IL-21R deficient MRL/*lpr* mice[34] to MRL/*lpr*-Igk^h/h to generate MRL/*lpr*-Igk^m/h-IL21R^−/− (IL-21R KO) mice.

In accordance with a previous report[34], we observed a reduction of B cell numbers in IL-21R-deficient MRL/*lpr*-Igk^m/h mice. However, only the FO B cell numbers were decreased while MZ B cells were not affected (Supplementary Fig. 6a, b). In the absence of IL-21R, the dual-κ FO and MZ B cells were still more activated than single-κ cells (Supplementary Fig. 6c), but their

**Fig. 2** Transcriptome and pathway analyses of FO and MZ single-κ and dual-κ B cells (**a**) RNASeq analysis of FO (left) and MZ (right) single and dual-κ B cells, sorted from spleen cells pooled from $N = 2$–4 mice for each biological replicate. Sorting strategy is described in Supplementary Fig. 2a. Heat maps display differentially expressed genes (FDR ≤ 0.05) between single-κ and dual-κ B cells of FO (1938 genes) or MZ (446 genes) cell subsets. The maps represent min/max row-scaled rlog values, where the expression values are mapped to colors using the minimum and maximum of each row/gene independently. **b** Differential expression of *Cd19*, *Cd79A*, and *IgκV* genes from RNAseq data described in **a** displayed as mean log2-fold change ± SEM in dual over single-κ cells for each group. The values for *IgκV* genes were calculated as the mean log2-fold change (±SEM) of all *IgκV* genes with FDR ≤ 0.05, which were $N = 14$ in FO and $N = 30$ in MZ cells (represented by symbols on the graph). A log2 value of zero indicates equal expression, while a value of 1 indicates a 2-fold difference. **c** Differential expression (mean for each group of the log2-fold change ± SEM) of *Jκ* genes in FO or MZ dual-κ relative to single-κ B cells. **d** B cell-relevant pathways identified by Ingenuity Pathway Analysis (IPA) as differently activated (based on FDR ≤ 0.1) in FO (top) and MZ (bottom) dual-κ B cells relative to single-κ B cells. Vertical dashed lines indicate the 0.05 -log(p-value) above which the pathway is considered significantly enriched. Data analysis was performed using DESeq2 and significance was assessed by the Wald test with the Benjamini and Hochberg adjustment. Data in all panels are from three independent biological replicates analyzed in one experiment

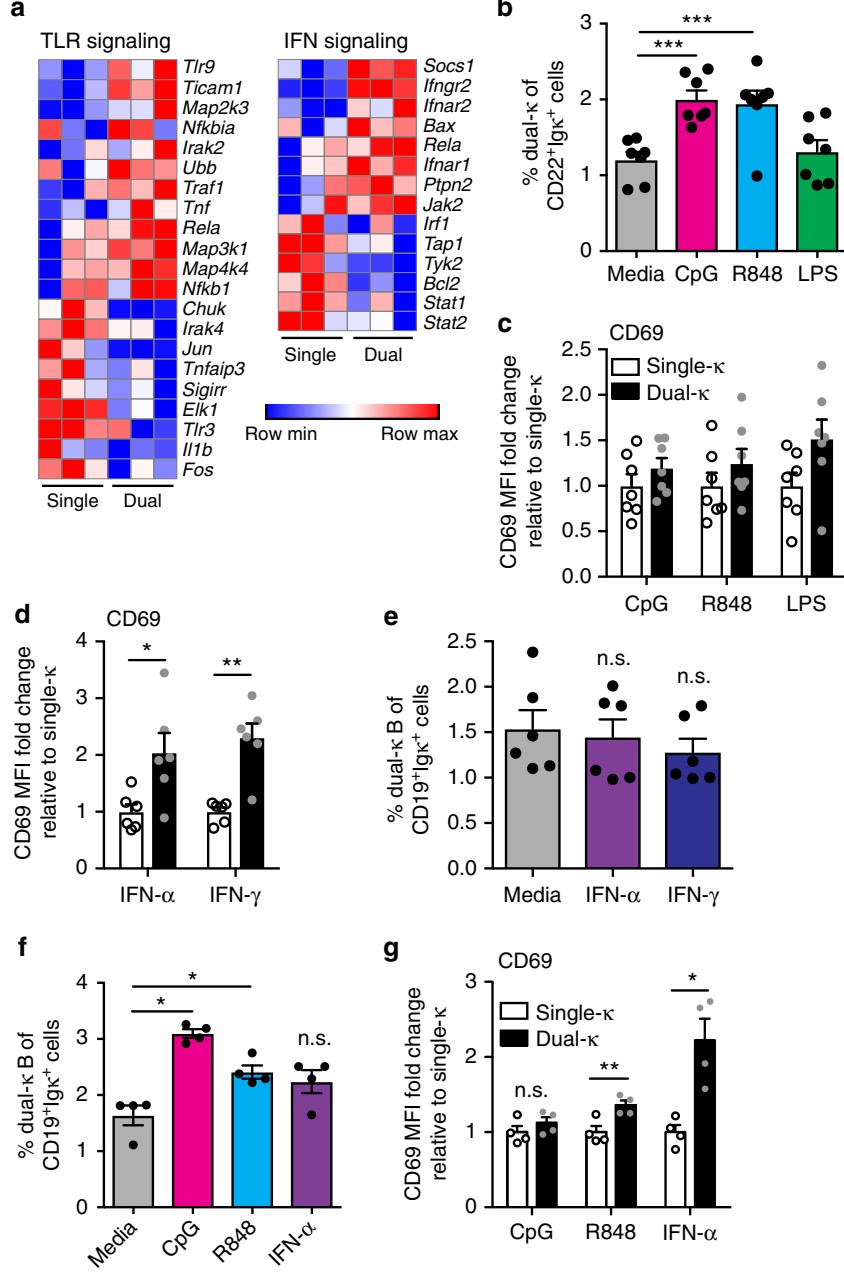

**Fig. 3** Innate stimuli expand and activate dual-κ B cells. **a** Relative expression of genes in dual-κ and single-κ FO B cells in the Toll-like Receptor (TLR, left; differential expression of 21 out of 76 genes) or Interferon (IFN, right; differential expression of 14 out of 36 genes) signaling pathways identified in the RNAseq data described in Fig. 2 (FDR ≤ 0.1). The heat maps colors are explained in Fig. 2a. Data are from three independent biological replicates analyzed in one experiment. **b, c** Mean frequency of dual-κ B cells (**b**) and fold change of CD69 expression (MFI) on dual-κ relative to single-κ cells (**c**) within CD43⁻ B cells cultured for 48 h in media or with CpG for TLR9, R848 for TLR7, LPS for TLR4. Gating strategy for this analysis and subtraction of background events are described in Supplementary Fig. 3a, b. Data are combined from two independent experiments using 14–15-wk-old MRL/*lpr-Igk^{m/h}* mice (N = 7 total). Data in **c** are not significant. **d, e** Untouched (CD43⁻) B cells were stimulated for 24 h with IFNα or IFNγ. Bar graphs show MFI of CD69 on dual-κ cells as a fold change relative to amounts on single-κ cells (**d**), or the mean frequency of dual-κ B cells (**e**), within the CD19⁺Igκ⁺ cell population gated as in Supplementary Fig. 3a. Data are combined from two independent experiments (N = 6 total) using 12-wk-old MRL/*lpr-Igk^{m/h}* mice. **f, g** Untouched naive B cells (magnetic selection of CD43⁻CD80⁻CD86⁻ cells, Supplementary Fig. 4c) were stimulated for 48 h with CpG, R848, or IFNα. Shown are the mean frequencies of dual-κ B cells (**f**) or the MFI of CD69 (**g**) within the CD19⁺Igκ⁺ cells gated as in Supplementary Fig. 3a. Data were obtained using 12-wk-old MRL/*lpr-Igk^{m/h}* mice (N = 4) in one experiment. In panels (**c, d–g**), the CD69 MFI values of cells cultured in media were subtracted from those in stimulated samples. *P < 0.05, **P < 0.01, ***P < 0.001; n.s. not significant. Significance was assessed by Student's t- or Mann–Whitney tests. In all bar graphs the standard error bars represent SEM and each symbol an individual mouse

frequency was significantly decreased (Fig. 5a, b). This reduction was already present at the immature/transitional B cell stage in the spleen (Fig. 5c), suggesting that the generation and/or initial expansion of immature dual-κ B cells depends on IL-21.

In agreement with previous studies[34], the total numbers of CD138⁺CD44^{high} plasmablasts were 2–3-fold reduced in IL-21R KO mice (Supplementary Fig. 7a, b), but this did not equally affect single and dual-κ cells. In fact, there was a two-fold

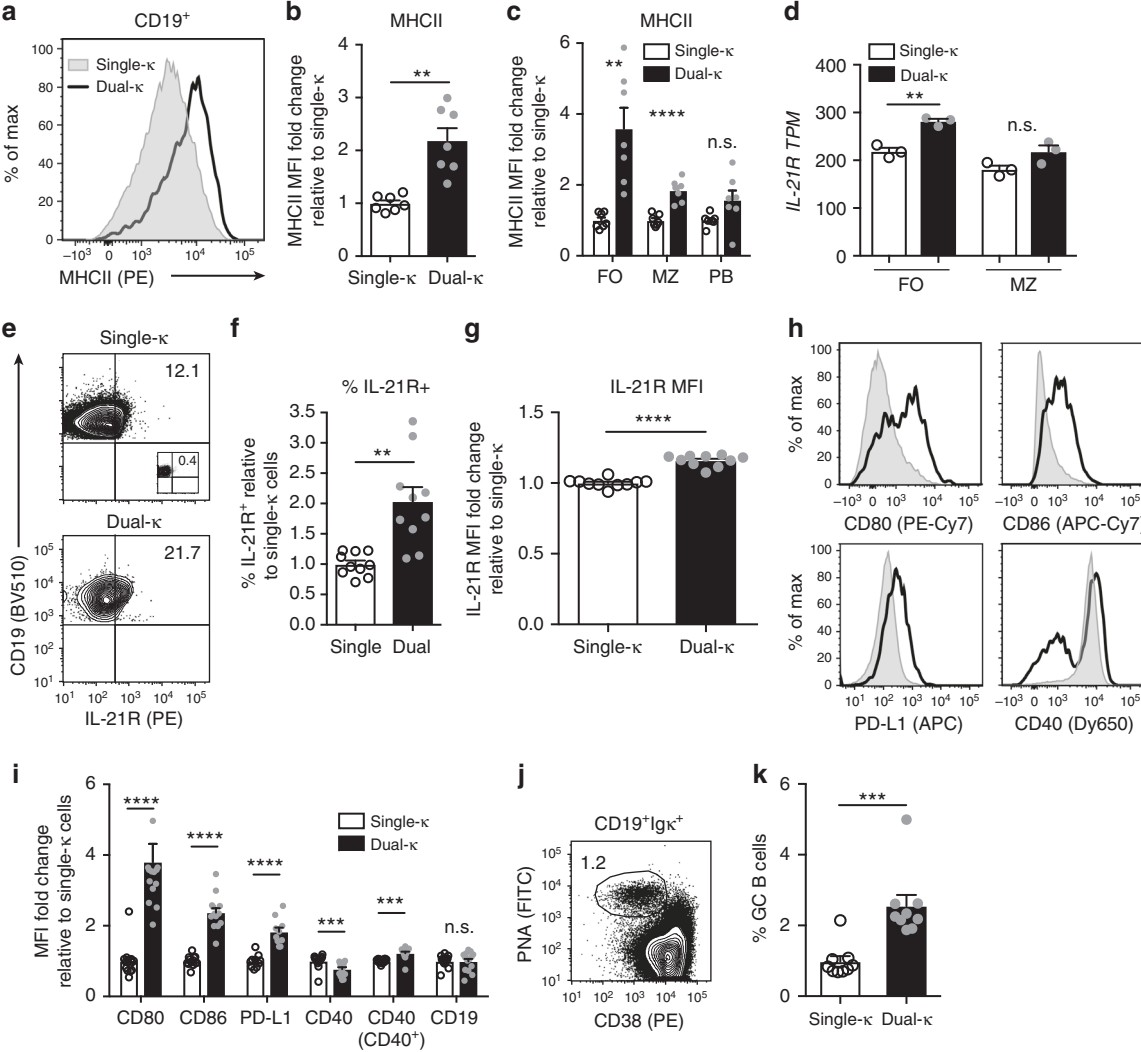

**Fig. 4** Dual-κ B cells upregulate several T cell co-stimulatory receptors. **a** Representative MHCII I-A$^k$ expression on single-κ (shaded gray) or dual-κ (black line) CD19$^+$ B cells gated as in Supplementary Fig. 1a, b. **b** Quantification of MHCII described in **a**. **c** MHCII expression on dual-κ relative to single-κ FO and MZ B cells (gated as in Supplementary Fig. 6a), and plasmablasts (PB, gated as in Supplementary Fig. 1a, b). Data in **b**, **c** are pooled from two independent experiments (N = 7 total) using 13–15-wk-old MRL/*lpr-Igk*$^{m/h}$ mice. **d** *IL-21R* transcripts per million (TPM) in single or dual-κ FO and MZ B cells analyzed by RNAseq (described in Fig. 2). Data are from three independent biological replicates analyzed in one experiment. **e** Representative analysis of IL-21R expression on CD19$^+$ single-κ or dual-κ B cells (gated as in Supplementary Fig. 1a, b) showing frequencies of IL-21R$^+$ cells. IL-21R knockout splenocytes (inset plot) were used to set gates. **f** Frequencies of IL-21R$^+$ cells as fold change relative to single-κ cells, analyzed as described in panel (**e**). **g** Relative MFI of IL-21R on IL-21R$^+$ B cells from analyses in **e**, **f**. Data in **f**, **g** were combined from three independent experiments (N = 10 13–14-wk-old MRL/*lpr-Igk*$^{m/h}$ mice). **h** Representative analyses of indicated co-receptors on CD19$^+$ single-κ (shaded gray) or dual-κ (black line) B cells gated as in Supplementary Fig. 5a, top row. **i** Expression (MFI) of indicated surface proteins analyzed as in panel (**h**) on dual-κ cells as a fold change relative to single-κ cells. CD40 is shown for total or CD40$^+$ cells. Data were combined from two independent experiments (N = 10–14 13–16-wk-old MRL/*lpr-Igk*$^{m/h}$ mice). One CD80 dual-κ point (10.05 value) is outside the scale. **j** Representative analysis of MRL/*lpr-Igk*$^{m/h}$ GC B cells within CD19$^+$Igκ$^+$ cells gated as in Supplementary Fig. 1a, b. **k** Frequency of GC single or dual-κ B cells described in panel (**j**). Data were combined from three independent experiments (N = 9). **P < 0.01, ***P < 0.001, ****P < 0.0001; n.s., not significant. Significance was assessed by Student's t- or Mann–Whitney tests. Bar graphs report SEM and symbols represent individual mice (besides in panel **d**)

reduction in the relative frequency of dual-κ cells in the plasmablast compartment of IL-21R-deficient MRL/*lpr-Igk*$^{m/h}$ mice (Fig. 5d, e). To investigate whether IL-21R signaling confers the enhanced proliferation observed in dual-κ B cells (Fig. 1d), we injected IL-21R sufficient and deficient animals with EdU and analyzed cells 24 h later. Although dual-κ B cells displayed more proliferation than single-κ B cells even in the absence of IL-21R signaling, the difference was much smaller, and the extent of proliferation was severely reduced, when compared to IL-21R sufficient animals (Fig. 5f).

The contribution of IL-21 to the generation of IgG$^+$ B cells is controversial, with reports indicating reduced, similar, or enhanced IgG$^+$ memory B cells in IL-21R KO mice[32,33]. We found that the total numbers of IgG$^+$, κ$^+$ memory B cells were similar in MRL/*lpr* with or without IL-21R ablation, though trended to higher numbers in KO mice (Supplementary Fig. 7c, d). Nevertheless, the frequency of dual-κ cells within the IgG$^+$CD138$^-$ B cell compartment was reduced by about 2–3-fold in IL-21R KO mice (Fig. 5g, h). Given that the FO compartment is a major precursor population for both

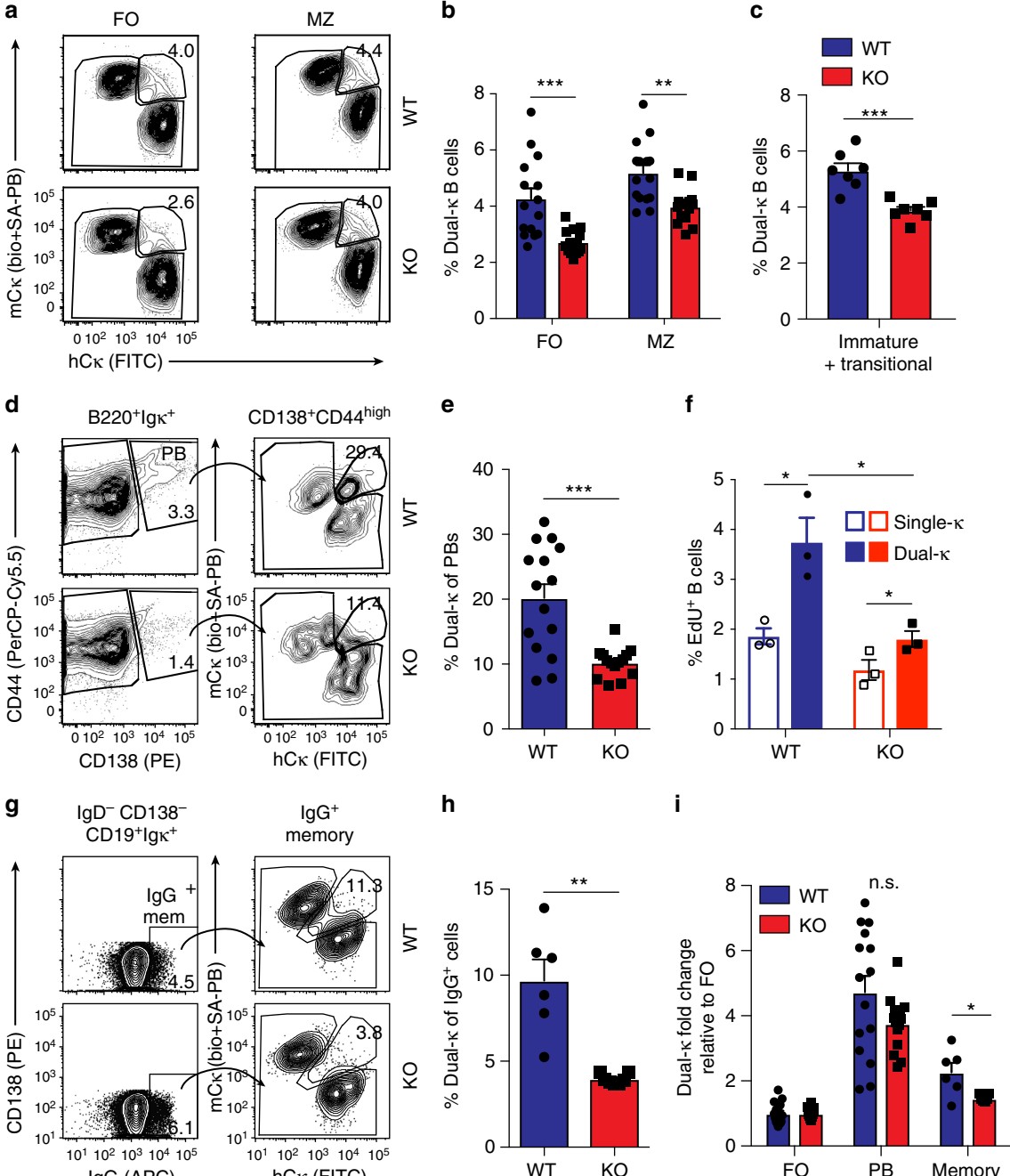

**Fig. 5** Contribution of IL-21R signaling to the expansion and differentiation of dual-κ B cells. **a** Representative analysis of single and dual-κ B cells in FO and MZ subsets of wt and IL-21R KO MRL/*lpr-Igk*^*m/h* mice. Cell subsets were gated as shown in Supplementary Fig. 6a. **b** Mean frequencies of dual-κ B cells in FO or MZ κ⁺ subsets described in **a**. Data are combined from five independent experiments with *N* = 15 wt and 14 KO mice. **c** Mean frequency of dual-κ cells within CD3⁻CD19⁺CD21⁻/lowCD24high splenic immature/transitional B cells (gated as in Supplementary Fig. 4a, SP), combined from two independent experiments (*N* = 7 for both wt and IL-21R KO MRL/*lpr-Igk*^*m/h* mice). **d** Analysis of CD138⁺CD44high single and dual-κ plasmablasts (PBs, gated as in Supplementary Fig. 7a) from splenocytes of wt and IL-21R KO MRL/*lpr-Igk*^*m/h* mice. **e** Mean frequency of dual-κ cells within the PB cell subset described in (**d**). Data are combined from five independent experiments with *N* = 15 wt and 14 KO mice. **f** EdU incorporation by splenic single-κ (open bar) or dual-κ (filled bar) CD3⁻B220⁺ B cells (gated as in Supplementary Fig. 1a, c), 24 h after EdU injection in 12–17-wk-old wt or IL-21R KO MRL/*lpr-Igk*^*m/h* mice (*N* = 3, in one experiment). **g** Analysis of CD138⁻IgG⁺ (IgG1,IgG2a/2b) single or dual-κ memory B cells (gated as in Supplementary Fig. 7c) in spleen of wt and IL-21R KO MRL/*lpr-Igk*^*m/h* mice. **h** Frequency of dual-κ cells within the memory IgG⁺ B cell subset shown in **g**. Data are combined from three independent experiments with *N* = 6 wt and 7 KO mice. **i** Fold change of dual-κ B cell frequencies in PBs and IgG⁺ cell subsets of wt and IL-21R KO MRL/*lpr-Igk*^*m/h* mice relative to those in their respective FO B cell subset, based on data in panels (**b**–**e**–**h**). Data in panels (**b**, **c**, **h**, **i**) were from 15–20-wk-old wt or IL-21R KO MRL/*lpr-Igk*^*m/h* mice. *P < 0.05, **P < 0.01, ***P < 0.001; n.s. not significant. Significance was assessed by Student's *t*- or Mann–Whitney tests. In all bar graphs the standard error bars represent SEM and each symbol represents an individual mouse

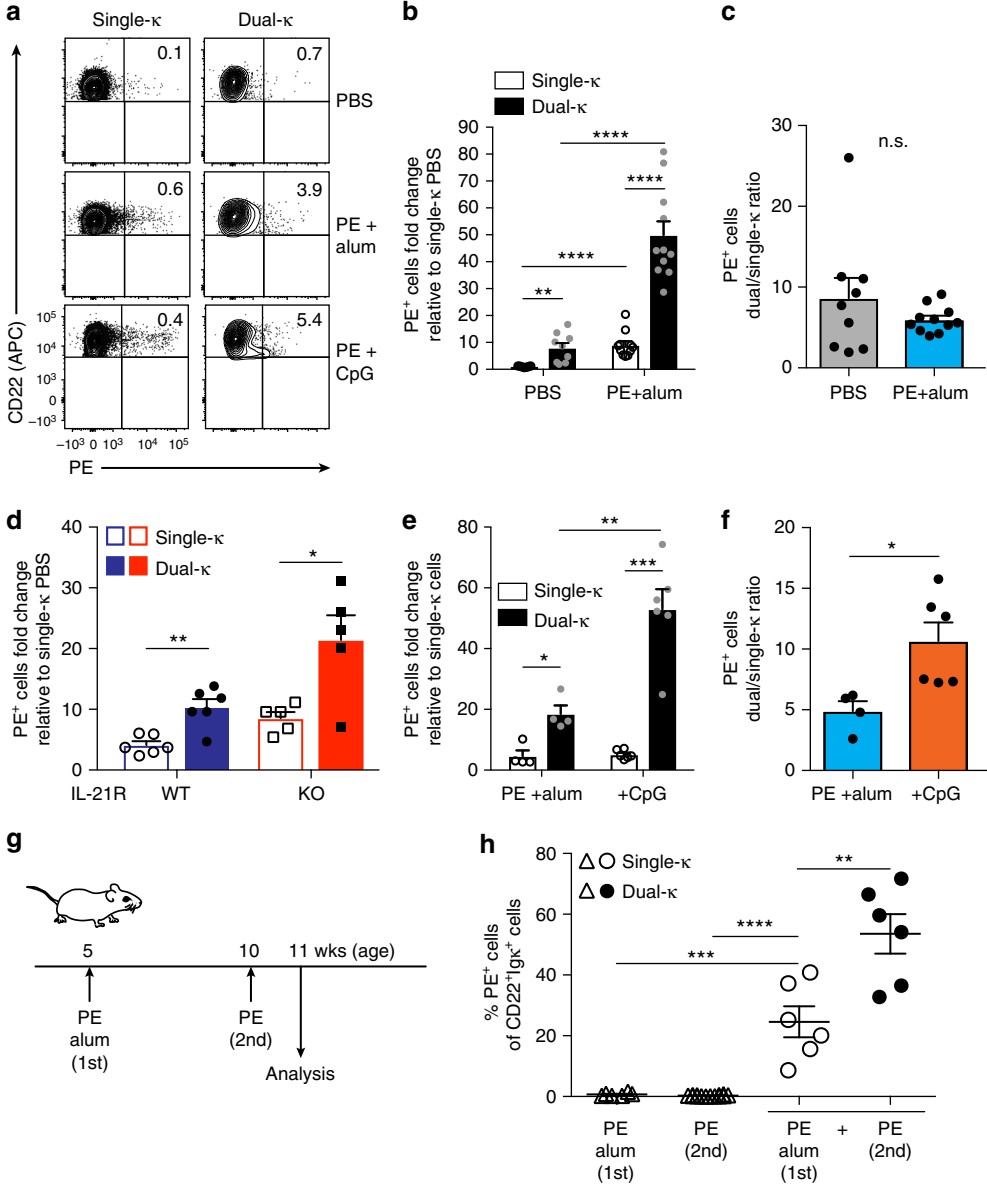

**Fig. 6** Primary and memory T cell-dependent responses of single-κ and dual-κ B cells. **a** Representative analysis of PE-binding single and dual-κ cells within CD22+Igκ+ splenocytes (gated as in Supplementary Fig. 8b) of MRL/*lpr-Igk*^m/h^ mice 7 days after injection of PBS, PE + Alum, or PE + CpG. **b** Frequencies of PE-binding single and dual-κ cells (analyzed as in panel **a**) from MRL/*lpr-Igk*^m/h^ mice, 7 days after injection of PBS or PE + Alum. Data represent fold changes relative to PE-reactive single-κ B cells in PBS-injected mice. **c** Ratio of dual-κ to single-κ PE-binding cells in MRL/*lpr-Igk*^m/h^ mice treated as indicated. Data in (**b**, **c**) are combined from four independent experiments with 6–7-wk-old MRL/*lpr-Igk*^m/h^ mice injected with either PBS (N = 9) or PE + Alum (N = 11). **d** Frequencies of PE-binding single and dual-κ cells (analyzed as in panel **a**) from wt (N = 6) and IL-21R KO MRL/*lpr-Igk*^m/h^ (N = 5) mice 7 days after injection of PE + Alum, shown as fold change relative to PE-reactive single-κ B cells in naive mice of corresponding strain (N = 3 and N = 6, respectively, combined from two experiments). **e** Frequencies of PE-binding single or dual-κ B cells (analyzed as in panel **a**) from MRL/*lpr-Igk*^m/h^ mice immunized with either PE + Alum or PE + CpG, shown as fold change relative to single-κ B cells in PBS-injected mice. **f** Ratio of dual-κ to single-κ B cells in mice immunized as indicated. Data in (**e**, **f**) are combined from two independent experiments using 6–12-wk-old MRL/*lpr-Igk*^m/h^ mice immunized with PE + Alum (N = 4 total) or PE + CpG (N = 6 total). **g** Schematics of experiment to measure memory response of PE-binding B cells. **h** Frequencies of PE-binding single or dual-κ B cells (analyzed as in panel **a**) from control MRL/*lpr-Igk*^m/h^ mice receiving only a primary (PE + Alum, N = 3) or secondary (PE, N = 7) immunization, and experimental mice receiving both primary and secondary immunizations (N = 6), combined from three independent experiments. Single and dual-κ cells for control mice are shown together. *P < 0.05, **P < 0.01, ***P < 0.001, ****P < 0.0001; n.s. not significant. Significance was assessed by Student's *t*- or Mann–Whitney tests. In all bar graphs the standard error bars represent SEM and each symbol represents an individual mouse

plasmablasts and memory B cells, we questioned whether the reduced frequency of dual-κ cells in the effector populations of IL-21R KO mice was due to the initial effect observed in the FO pool. To gauge this, we normalized the frequencies of dual-κ cells in both effector populations to that of the respective FO B cell population in each mouse. Once normalized, we observed a significant reduction in the expansion of IL-21R KO dual-κ B cells into the memory B cell pool, but not into the plasmablast compartment (Fig. 5i). This suggests that IL-21 plays an additional, or more prominent, role in the generation of dual-κ

memory B cells independent of its effect in establishing the FO B cell pool.

Overall, these data indicate that IL-21 signaling significantly contributes to the entry and enrichment of dual-κ B cells in the naive (FO and MZ), blasting, and memory B cell subsets of MRL/*lpr* mice.

**Enhanced T cell-dependent response of dual-κ cells.** Given dual-κ B cells overexpress many co-stimulatory receptors used for cognate interaction with T cells, we next investigated whether these cells mount higher T cell-dependent antigen-specific responses. To address this, we compared the response of single-κ and dual-κ B cells to phycoerythrin (PE), a well characterized T cell-dependent protein antigen model that allows relative straightforward identification of antigen-reactive B cells by flow cytometry[35].

Because MRL/*lpr* mice fail to respond to T cell-dependent antigens as disease progresses[36], we immunized young (6–7-wk-old) MRL/*lpr-Igk^{m/h}* mice with PE in Alum confirming that the PE-specific IgG antibody response is comparable in magnitude to that of non-autoimmune CB17-*Igk^{m/h}* mice (Supplementary Fig. 8a). At day 7 after PE/Alum immunization, the frequency of PE-binding single-κ and dual-κ B cells within each MRL/*lpr-Igk^{m/h}* mouse was analyzed using flow cytometry as a measure of their response to antigen (Fig. 6a and Supplementary Fig. 8b). To compare the responses in different experiments, we normalized the frequency of PE-responding single and dual-κ B cells in immunized mice to that of single-κ B cells in PBS-injected mice within each experiment. On average, the frequency of PE-reactive single-κ B cells increased almost 10-fold following PE/Alum immunization (Fig. 6b). For dual-κ B cells, this difference was >40-fold, suggesting an increased participation of this subset in the response to PE (Fig. 6b). However, we noticed that in naive (PBS-treated) animals, the frequency of PE-binding B cells (the antigen-specific clonal precursor frequency) was already higher in the dual-κ than the single-κ cell subset (Fig. 6a, b). In fact, the ratio of dual-κ to single-κ PE-reactive B cells did not change with PE/Alum immunization (Fig. 6c), indicating a similar ability of single and dual-κ B cells to expand during a primary T cell-dependent antigen response.

As shown above (Fig. 5d–f), IL-21-R signaling contributes to the proliferation of dual-κ B cells and their enrichment into the plasmablast subset in vivo. Thus, we next asked whether the expansion of PE-reactive dual-κ B cells was dependent on IL-21. To address this, we immunized groups of IL-21R KO and WT MRL/*lpr-Igk^{m/h}* mice with PE/Alum and compared the expansion of PE-reactive dual-κ B cells to that of single-κ B cells in naive mice from related strains. In the absence of IL-21R, both single-κ and dual-κ B cells were able to expand in response to PE (Fig. 6d), and this expansion was similar (4–6-fold) in all populations relative to their precursor frequency in naive mice, negating a role for IL-21 in this process.

Because dual-κ B cells display enhanced response to both TLR7 and TLR9 stimulation in vitro, we next investigated if TLR9 signaling increases the T cell-dependent response of dual-κ B cells in vivo. Hence, MRL/*lpr-Igk^{m/h}* mice were immunized with PE either in Alum or with the addition of CpG (Fig. 6a). As previously observed, there was a larger frequency of PE-binding cells within the dual-κ cell population following PE/Alum immunization (Fig. 6e). However, in mice that received PE with CpG, there was an even greater expansion of PE-reactive dual-κ B cells (Fig. 6e), which was doubled relative to single-κ cells (Fig. 6f).

Finally, we next questioned whether the larger frequency of antigen-reactive dual-κ B cells that follows a primary response to a T cell-dependent antigen leads to a superior memory response. Thus, we immunized 5 week-old MRL/*lpr-Igk^{m/h}* mice with PE/

Alum and, after 5 weeks, boosted these mice with PE without adjuvant and analyzed PE-reactive B cells one week later (Fig. 6g). The response of memory B cells was demonstrated by a much higher frequency of PE-binding cells in mice that received both primary and boost immunizations relative to those that received only primary or boost injections (Fig. 6h). In mice subjected to the memory protocol, the frequency of dual-κ B cells binding PE was almost 2-fold higher than that of single-κ B cells (Fig. 6h), denoting a greater ability of dual-κ memory B cells to respond to a recall antigen.

Together, these data indicate that independent of IL-21, dual-κ B cells clonally expand to a higher degree in response to a T cell-dependent antigen relative to single-κ B cells. This increased clonal expansion is partly due to a larger antigen-reactive clonal frequency in the naive population and partly to enhanced response to TLR agonists, when available. Furthermore, after a primary antigen encounter, dual-κ B cells differentiate into memory B cells that display heightened response to a secondary antigen encounter.

## Discussion

SLE affects about 0.2% of the world's population and has no available cure[37] and treatments are general immunosuppressants that have minimal specificity and unwarranted side effects. B cells represent a notable target for disease control, but current B cell targeted therapies promote immunodeficiency and increase risk to infections because they do not discriminate between B cells promoting SLE vs. those protecting from pathogens. Defining the characteristics of B cell subsets that specifically contribute to SLE pathogenesis, therefore, can lead to the development of therapies that ameliorate autoimmunity while preserving immunological defense. We and others have previously shown that B cells that co-express two distinct IgL chains and thus two BCRs (B$_{2R}$ cells) are significantly expanded in lupus-prone MRL and MRL/*lpr* mice[23] but not in NZB/NZW mice[25]. In the present work we indicate that T cell-dependent signals and innate stimuli, such as IL-21, type I and II IFN, and TLR agonists, all play an important role in the activation, expansion, and effector function of B$_{2R}$ cells in the MRL/*lpr* autoimmune background.

In addition to secreting antibodies, B cells function as antigen-presenting cells and, as such, can be involved in the activation of pathogenic T cells in diverse autoimmune diseases[34,38–41]. We show that dual-κ B cells of MRL/*lpr* mice express much higher levels of MHCII, suggesting they may be better able to present antigen to CD4 T cells. This is of interest in the context of murine lupus, as recent work has shown decreased T cell activation and disease symptoms in MRL/*lpr* mice bearing MHCII-deficient B cells[42]. Dual-κ B cells express greater amounts of additional surface receptors important for cognate engagement with T cells. Among these co-receptors we were intrigued by IL-21R, as IL-21 is an important T cell-derived soluble factor that aids the generation of GC B cells, Ig class switched plasma cells and, potentially, memory B cells[32,33]. Moreover, IL-21 blockade and ablation of IL-21 signaling in lupus-prone mice ameliorate disease symptoms by reducing total B cells, plasmablasts and autoantibody titers[34,43,44]. We show in our study that deficiency in IL-21R signaling reduces the frequency of dual-κ B cells in the naive (FO and MZ), the plasmablast and the IgG$^+$ memory B cell subsets of MRL/*lpr* mice, indicating dual-κ B cells have an enhanced dependency on IL-21 for their maturation. The reduction of dual-κ B cells in different subsets may have different explanations. For instance, dual-κ B cells were already decreased in the newly generated (immature and transitional) splenic B cell fractions of IL-21R KO mice, possibly because IL-21R is expressed during bone marrow B cell development and plays a role in the

generation of immature/transitional B cells[45]. Thus, the reduced frequency of dual-κ B cells in the naive mature B cell population of IL-21R KO mice may be simply the result of an increased dependency of dual-κ B cells on IL-21 during their earlier development. In regard to plasmablasts, the two-fold reduction present in IL-21R KO mice was equivalent to that observed in the naive FO B cell compartment leading us to suspect that these events may be connected. However, there could be alternative explanations. Based on reduced EdU incorporation, dual-κ B cells are less proliferative in IL-21R KO than wild-type mice, possibly leading to reduced blasting and, thus, plasmablast formation. Furthermore, IL-21R signaling in B cells is known to promote the generation of plasma cells in synergy with CD40 signaling[32,33], both of which, we show, are amplified in dual-κ (relative to single-κ) FO B cells. These observations suggest that dual-κ cells may have a better ability to differentiate into plasmablasts in response to IL-21 and CD40. The significant reduction of dual-κ memory B cells in IL-21R-deficient mice is of particular interest since we otherwise observed an upward trend, though not significant, of total memory B cells in the absence of IL-21 signaling. Dual-κ B cells were decreased in the memory compartment of IL-21R KO mice to a greater extent relative to the initial effect observed in FO B cells, leading us to conclude that the enrichment of dual-κ B cells into the memory cell subset is crucially dependent on IL-21R signaling. Presently, we cannot exclude that any of the differences observed in the frequency of dual-κ B cells in IL-21R KO mice are B cell-extrinsic and/or due to the absence of autoimmunity in these mice.

Dual-κ B cells also display a two-to-three fold higher expression of CD80, CD86, and PD-L1, which are co-receptors that promote $T_{FH}$ help and GC B cell survival[46,47]. Indeed, the frequencies of GC B cells, cMyc[+] cells, and proliferating cells were higher within the dual-κ cell subset, suggesting more frequent antigen responses and T cell help. Interestingly, this was not apparent during the initial B cell response to PE immunization. In fact, immunizations with the T cell-dependent antigen PE in Alum elicited the same magnitude of clonal expansion by both single and dual-κ B cells, and regardless of IL-21R signaling. Despite a similar expansion, the response of dual-κ B cells to the PE protein antigen was amplified relative to that of single-κ B cells, but this amplification was accounted for by an increased antigen-reactive clonal frequency in the naive dual-κ B cell population. This difference might be due to the expression of two Ig H + L chain combinations in each dual-κ cell, which doubles the chances of reactivity with any antigen, or to the increased expression of autoreactive and cross-reactive specificities, which has been reported[21,23]. It is likely that the system we employed to investigate B cell response to antigen, which is the frequency of cells binding antigen at day 7, does not reflect many of the effects mediated by T cell help, such as increased generation and survival of GC B cells, plasmablasts, and memory B cells. Indeed, when we compared single and dual-κ B cells during a memory response to PE, we observed a much larger expansion (i.e., response) of dual-κ cells after boosting. This strongly indicates that during the primary response dual-κ B cells differentiate into memory B cells in larger numbers and possibly because of enhanced T cell help.

The activation and differentiation of autoreactive B cells can also occur independent of T cell help[2]. Toll-like receptor signaling, particularly endosomal TLR7/TLR9, is required for the production of anti-nuclear antibodies and the amplification of (auto)Ab responses[48–50]. IPA together with in vitro studies indicated a differential ability of dual-κ B cells to respond to TLR7/TLR9 agonists. Indeed, we show that immunization of PE with the TLR9 agonist CpG leads to a two-fold greater expansion of PE-reactive dual-κ B cells in vivo relative to that elicited by Alum or in single-κ B cells, corroborating the increased

expansion to TLR7&9 agonists observed in vitro. Why dual-κ B cells display enhanced activity of the TLR7&9 pathways is unclear, but we speculate this may be related to the larger frequency of dual-κ cells binding nuclear antigens[23]. Indeed, the continual internalization of nuclear self-antigens would be expected to lead to an enhanced stimulation and activity of TLR7&9 signaling pathways. The IPA also revealed an elevated activity of IFN I and II signaling pathways in dual-κ B cells, which was confirmed by finding a larger frequency of activated dual-κ (than single-κ) cells in B cell cultures treated with IFN-α or IFN-γ. Gene expression studies have identified a type I IFN signature in SLE patients[51,52], and this cytokine exacerbates autoimmunity also in mice[53], where it can sensitize B cells to TLR7&9 signaling[54,55]. Moreover, recent studies have demonstrated that B cell-intrinsic IFN-γ receptor signaling promotes spontaneous GCs, autoantibodies, and systemic autoimmunity in two lupus-prone mouse strains[54,56] and potentially in humans[54]. We propose that these distinct factors, stronger TLR and IFN signaling pathways, together with a higher frequency of reactivity for nuclear antigens, play a major role in the enrichment of (autoreactive) dual-κ B cells in the mature naive, GC, plasmablasts and memory B cell compartments in mice with lupus.

Our RNAseq analyses were performed on B cells displaying a FO (CD21[+]CD21[low]CD24[low]) or a MZ (CD21[high]CD1d[high]) phenotype. It may be argued that the results of the FO cell analyses could have been partially affected by the presence of memory B cells and plasmablasts, which are in larger numbers in the dual-κ cell subset, and could display a FO phenotype. However, our data does not support this argument. In fact, although the FO B cell samples were likely contaminated by memory B cells and plasmablasts, the analysis of IgG and CD138 transcripts does not support a larger contamination in the dual-κ cell samples (Supplementary Fig. 2b). Furthermore, when we cultured naive (CD43[−]CD80[−]CD86[−]) B cells with TLR7&9 agonists or IFN-α, the responses of dual-κ cells were still above those of single-κ cells.

The transcriptome analysis revealed that the expression of one vs. two different BCRs on the surface of MRL/*lpr* B cells results in a differential gene expression of many molecules involved in BCR signaling, especially in FO B cells. The three main MAPK signaling pathways (ERK, JNK, p38), which were differentially enriched in dual-κ over single-κ FO B cells, have been shown to be activated in B cells of SLE patients[57] and of mouse models of lupus[58]. IPA also identified a differential activation of the NF-κB signaling cascade, a pathway whose activation in B cells has been associated with SLE[58–60] and that spreads downstream of both TLR and CD40. Other signaling pathways that have been reported to be highly activated in B cells of lupus mice are the PI3K and mTOR[58], which were also enriched in dual-κ B cells. Activation of PI3K has been shown to break peripheral tolerance in anergic autoreactive B cells leading to their activation and response[61], and could similarly support the preferential expansion of autoreactive dual-κ B cell clones. We assume that the observed transcriptome differences in the activity of many signaling pathways between single- and dual-κ B cells reflect, at least to a degree, increased self-antigen-mediated signaling by an autoreactive BCR. Interestingly, dual-κ B cells show a higher degree of activation (CD80 and CD86 expression) already in immature and transitional B cell subsets. The markers we used to define immature/transitional B cells (lower CD23, IgD, and CD21, and higher CD24) might also capture a small fraction of activated mature B cells. However, this fraction would be likely too small compared to immature/transitional B cells to account for the observations we describe. Thus, we contend these findings indicate that dual-κ B cells distinguish from single-κ B cells

starting at their origin, and that this difference magnifies as disease and inflammation progress.

Recently, increased frequency of allelically included (κ+λ+) $B_{2R}$ cells has also been described in about half of SLE patients in the UK[26], elevating the clinical significance of our findings in mice. Thus, increased frequency of $B_{2R}$ cells may be a marker of disease in a large subset of lupus patients, and this B cell subset may one day represent a relevant cell target in lupus. While the dual-κ B cells present in MRL/lpr mice serve as an excellent model to investigate the mechanistic aspects of $B_{2R}$ cell function in lupus, this mouse model has its limitations as it does not take into account the genetic diversity that is present in the general human population. The MRL/lpr mouse strain also carries an inactivating mutation in Fas, which likely adds to the general B cell activation. Therefore, future studies will investigate if the molecular pathways and cell surface receptors differentially expressed in B cells of MRL/lpr mice are also present in $B_{2R}$ cells of autoimmune patients.

Collectively, our results show that dual-κ B cells in the MRL/lpr autoimmune background, which are cells that are mostly autoreactive[23], are also highly proliferative and activated, and they display a distinct gene signature that predisposes to a preferential response to innate stimuli and T cell-derived signals leading to enhanced enrichment and differentiation into effector B cells. These unique characteristics are those expected in key B cell players in autoimmune responses and explain the advantage dual-κ B cells display over single-κ B cells during lupus disease in MRL/lpr mice. They also reveal potential targets for their selective ablation in autoimmunity.

## Methods

**Mice.** $Igk^{m/h}$ in the MRL/lpr and CB17 genetic backgrounds have been previously described[22,23]. The CB17 background of CB17-$Igk^{m/h}$ mice is actually a mixed CB17 x BL/6 × 129 genetic background. MRL/lpr-IL21R$^{-/-}$ mice[34] were bred to MRL/lpr-$Igk^{h/h}$ to generate MRL/lpr-$Igk^{m/h}$-IL21R$^{-/-}$ mice. Both males and females 6–20-wk-of-age were used in this study. Animals were randomly assigned to groups based on availability, but considering their age and sex. Age was considered for severity of disease: disease symptoms start around 10 weeks of age and increase with age. Disease in males is slightly delayed in MRL/lpr mice and, thus, males assigned to groups were generally about two weeks older than females in same groups. For immunization studies, younger mice were used because older MRL/lpr mice do not respond to vaccination. The age of the animals used in analyses is reported in figure legends. No animals were excluded from the analyses besides one MRL/lpr-IL21R$^{-/-}$ mouse that showed absolutely no response to PE immunization (i.e., frequency of PE+ B cells was in the range of naive mice): based on our experience this was due to a failure of i.p. injection. Group sizes were chosen based upon knowledge of the variation of individual analyses and to maximize the chances of uncovering statistically significant differences of the mean. All mice were housed in Specific Pathogen Free (SPF) conditions. All animal procedures were approved by the NJH (AS2533) and UCD-AMC (B-105314(05)1E and B-105317(03)1E) Institutional Animal Care and Use Committees and carried out in accordance with approved guidelines.

**Cell preparation.** Single-cell suspension from spleen were prepared in RPMI (Gibco) supplemented with 3% FBS. To remove erythrocytes cells were incubated for 2 min with a buffer composed of 0.15 M $NH_4Cl_4$, 10 mM $KHCO_3$, and 0.1 mM EDTA (pH 8.0). Cells were then centrifuged at 300× g and washed with media.

**Antibodies and flow cytometric analysis.** Single cells isolated from bone marrow and spleen tissues were first incubated with a rat anti-mouse FcγRII/III antibody (2.4G2; homemade) in FACS buffer (PBS 1X with 3% FBS and 0.05% $NaN_3$) for 15 min on ice to prevent unspecific binding of staining antibodies on cells through their Fc portion. Cells (2–3 × 10^6 for each staining panel) were then incubated for 20 min on ice in 50 μl FACS buffer that included antibodies against surface markers, starting with biotin-conjugated antibodies in a first step staining and then with a mix of fluorochrome-conjugated antibodies and streptavidin in a second step staining. Cells were washed with FACS buffer before and after each step. Fluorescent monoclonal antibodies against B220 (clone RA3–6B2; fluorochromes PE-Cy7 and APC-Cy7 at dilution 1:200 or PerCP-Cy5.5 at 1:100), CD1d (1B1; PE at 1:500 or PerCP-Cy5.5 at 1:100), CD19 (1D3; BV510 or APC; 1:200), CD138 (281–2; PE; 1:200), CD21 (7E9; APC; 1:200), CD22 (OX-97; APC; 1:500), CD23 (B3B4; BV786; 1:200), CD24 (M1/69; AF700 or APC-Cy7; 1:200), CD3e (145-2C11; PE at 1:300; PE-Cy7 at 1:100; APC-Cy7 at 1:100), CD38 (90; PE; 1:500),

CD44 (IM7; PerCP-Cy5.5; 1:400), CD69 (H1.2F3; PE-Cy7; 1:200), CD80 (16-10A1; APC or BV605; 1:200), CD86 (GL1; APC or APC-Cy7; 1:200), IgD (11–26 c.2a; BUV395 or APC-Cy7; 1:200), IL-21R (4A9; PE; 1:200), and PD-L1 (10 F.9G2; APC; 1:200) were purchased from BioLegend. Antibodies to MHCII I-A$^k$ (11–5.2; PE; 1:400), IgG1 (A85-1; APC; 1:100), and IgG2a/2b (R2-40; conjugated in our lab to Dylight 650 ThermoFisher; 1:100) were purchased from BD. Goat polyclonal Fab fragment antibodies to mouse IgM (cat. #115-097-020; FITC; 1:200) were purchased from Jackson Labs. Antibodies to CD40 (clone 1C10; used at 1:200) were made and conjugated to Dylight 650 (ThermoFisher) in our lab. Staining for Igκ was performed using a fluorescent polyclonal goat Fab' anti-human Igκ (Protos Immunoresearch cat. #376; FITC; 1:400) or a biotin-conjugated rat Fab anti-mouse Igκ (187.1; used at 1:100) that was made in our lab[26]. Biotin-labeled antibodies were visualized with streptavidin (eBioscience/ThermoFisher; cat. # S11222 Pac-Blue; 1:200). Peanut agglutinin (PNA;FITC; 1:1000) for the staining of GC B cells was purchased from Vector Laboratories. Intracellular staining for Ki67 (BioLegend; clone 16A8; BV605; 1:50) was performed using the True-Nuclear Transcription Factor Buffer Set (BioLegend). To stain for c-Myc, cells were first fixed in 2% paraformaldehyde and then permeabilized in 90% methanol. Cells were washed in PBS and then stained with rabbit anti-mouse c-Myc (D84C12, Cell Signaling; 1:100) followed by polyclonal donkey anti-rabbit IgG (BioLegend; #406406; Dylight-649; 1:100). Total rabbit IgG (Southern Biotech; 1:2000) was used as an isotype control for c-Myc staining. To stain for PE-specific B cells, 20 × 10^6 splenocytes per sample were first incubated with 4 μg of PE (Prozyme) in 1 ml staining buffer (1% BSA, 2 mM EDTA) for 30 min on ice. Cells were then washed twice with FACS buffer and surface staining was continued as described above. Dead cells were excluded from analyses either by 7-amino-actinomycin D (7AAD; eBioscience) or propidium iodide (Sigma-Aldrich) incorporation, or based on forward and side scatter. Sample acquisition was done using an LSRII or LSRFortessa cytometers (BD) and analyzed with FlowJo software v10.4.1 (Tree Star). Data were analyzed on a live and lymphoid cell gate (based on forward and side scatter), followed by a doublet cell exclusion gate to eliminate cell aggregates and doublets. The lymphoid gate at times set to capture activated cells displaying higher FSC and SSC, as these cells are frequent in autoimmune mice. The CD3+B220+ T cell population that is characteristic of MRL/lpr mice was excluded from the analyses of B cells by either gating out CD3+ events, or by gating B cells as CD19+ or CD22+, or as B220+ in combination with Igκ+, depending on the panel.

**Sorting of single-κ and dual-κ B cells.** Splenocytes were pooled from N = 2–4 mice for each biological replicate. These cells were first depleted of CD3 cells by complement-mediated lysis in the following way. Splenocytes were incubated with anti-CD3 antibodies (145-2C11, homemade) at a concentration of 0.4 μg/10^6 cells for 20 min on ice. Cells were then washed and resuspended in media containing reconstituted Low-Tox Guinea Pig Complement (Cedarlane) at a 1:20 dilution and incubated for 30 min at 37 °C. Afterwards, the cells were washed twice with media and then incubated with a rat anti-mouse FcγRII/III antibody (2.4G2; homemade) in FACS buffer for 15 min on ice to prevent subsequent unspecific binding of staining antibodies. Cells were then stained with fluorochrome-labeled antibodies. Stained cells were sorted as single or dual-κ CD3−CD21+CD1d^low^CD24^low FO B cells or CD3−CD21^high^CD1d^high^ MZ B cells (as shown in Supplementary Fig. 2a). Sorting purity was generally >95% for FO B cells and >70% for MZ B cells in three independent sorting experiments, and numbers of sorted cells varied from 5 × 10^4 to 8 × 10^6 depending on the population (5 × 10^4–16 × 10^4 for dual-κ cells). Cell sorting was performed using an ICyte Synergy cell sorter (Sony).

**RNAseq and gene pathway analyses.** Three independently sorted populations of single-κ and dual-κ FO or MZ B cells (sorted as described above under Sorting) were used for RNAseq analyses. Total RNA was extracted using the RNeasy Micro Kit (Qiagen) and quality was determined by a BioAnalyzer (Agilent Technologies). The RNA Integrity Number (RIN) for all samples was greater than 8. The RNA was processed for next-generation sequencing (NGS) library construction as developed in the NJH Genomics Core Service facility for analysis with a Life Technologies (Carlsbad, CA, USA) Ion Proton NGS platform. A modified Clontech SMARTer® Ultra™ Low Input RNA Kit for Sequencing - v3 (Mountain View, CA, USA) and modified Kapa Biosystems KAPA Hyper Prep Kit (Wilmington, MA, USA) were used. Briefly, library construction started with 1 ng of RNA, followed by SMARTer 1st strand cDNA synthesis (Takara), full length dscDNA amplification by LD-PCR, followed by purification and validation. Afterward, the samples were sheared using a Covaris focused-ultrasonicator, concentrated by vacuum, and then processed through the Kapa protocol. The Kapa protocol consisted of end repair and A-tailing, adapter ligation, library amplification, and a post-amplification cleanup step. Once validated, the libraries were sequenced as barcoded-pooled samples on an Ion Torrent Proton Sequencer (ThermoFisher Scientific) using a P1 Ion Proton chip (vP1.1.17) and the Torrent Suite software v5.0.2. Reads greater than 30 bp were mapped to the mouse mm10 reference genome with STAR v2.4.1d[62]. Uniquely mapping reads were counted for Ensembl Release 78 gene annotations using HTSeq v0.6.0 in intersection-nonempty mode[63]. Statistical significance for the differential gene expression between single and dual-κ B cells was evaluated using DESeq2 package v1.4.5[64] for the R statistical software v3.2.0 (R Core Team, 2014, Vienna, Austria) using the Wald test. P-value adjustment was performed with the Benjamini and Hochberg false discovery rate method within DESeq2[65].

We used a statistical model with variables for the tissue of origin (FO or MZ), the receptor type (single- or dual-κ), and their interaction, as well as a variable for the three independently sorted splenocyte cell pools, to account for any batch effects. A FDR ≤ 0.05 was used to evaluate differential expression of individual genes. Differentially expressed genes were evaluated based on the contrast FO or MZ single-κ vs. dual-κ cells. Pathway analysis was performed with IPA, IPA (QIAGEN Inc., https://www.qiagenbioinformatics.com/products/ingenuity-pathway-analysis/). Genes with a FDR of ≤ 0.1 were selected as differentially expressed for IPA analysis. Heat maps were generated in Morpheus [https://software.broadinstitute.org/morpheus/] using min/max row-scaled rlog values, where the expression values are mapped to colors using the minimum and maximum of each row/gene independently.

**In vitro cell culture**. Spleen (untouched) B cells were enriched by negative selection with anti-CD43 beads (Miltenyi Biotech) on an AutoMACS Pro Separator following manufacturer instructions. Enriched B cells (92–99% pure) were cultured in complete RPMI at $2 \times 10^6$ cells/ml for 24 or 48 h and then analyzed by flow cytometry. Where appropriate, the cells were stimulated with 50 nM CpG ODN 1826 (IDT), 10 μg/ml LPS (Sigma-Aldrich), 1 μg/ml R848 (Invivogen), 100 U/ml recombinant mouse IFN-α11 (PBL Assay Science), 50 ng/ml IFNγ (R&D Systems), 50 ng/ml BAFF (R&D Systems), 50 ng/ml APRIL (PeproTech), or 15 μg/ml anti-CD40 antibodies (IC10, made in house). To isolate naive B cells for in vitro studies, splenocytes were stained with PE-labeled antibodies against CD80 (BD clone B7-1; 1:400) and CD86 (BD clone B7-2; 1:400) and then cells were negatively selected with a combination of anti-PE and anti-CD43 beads (Miltenyi Biotech). Purity of enriched naive B cells was about 99%. To control for non-specific dual-κ B detection, which was an issue with TLR stimulation, for each experimental condition we also cultured a 1:1 mixture of homozygous $Igk^{h/h}$ and wild-type $Igk^{m/m}$ (h + m) cells (Supplementary Fig. 3b) and these background events were subtracted from those measured in TLR-stimulated h/m cultures.

**Immunizations and ELISA**. For PE immunizations, mice were injected i.p. with 150 μg PE (Prozyme) mixed with either equal volume of Alu-Gel-S (Aluminum hydroxide, SERVA) or 50 μg CpG ODN 1826 (Invivogen) in PBS, and euthanized 7 d later for analysis. For memory studies, the mice were boosted 5 weeks after the initial immunization, with 150 μg of PE in the absence of adjuvant, and euthanized 7 d after the boost. To measure anti-PE IgG serum antibodies, 96-well Nunc Immuno MaxiSorp plates (Thermo Fisher Scientific) were coated with 5 μg/ml PE in PBS overnight at 4 °C. Plates were washed three times with PBS/0.5% Tween-20, blocked for 2 h at 37 °C with blocking buffer (1% BSA in PBS), then washed again. Serum samples were diluted starting at 1:50 in blocking buffer, then three-fold serial dilutions were made in the same buffer and plates were incubated for 2 h at 37 °C. Plates were washed, incubated with alkaline phosphatase-conjugated goat anti-mouse IgG antibodies (Southern Biotech; cat #1030-04) diluted 1:1000 in blocking buffer, for 1 h at 37 °C, then washed again. To develop plates, 1 mg/ml of alkaline phosphatase substrate (Sigma-Aldrich) diluted in developing buffer (1 M diethanolamine, 8.4 mM $MgCl_2$, and 0.02% $NaN_3$, pH 9.8) was added to the plates. Absorbance values were read at 405 nm with a VersaMax ELISA Reader (Molecular Devices).

**In vivo EdU proliferation studies**. Mice were injected i.p. with 1 mg EdU (Sigma-Aldrich) in PBS and euthanized 24 h later. Spleen cells were harvested from EdU treated and untreated mice and stained for surface markers with antibodies listed above under "Antibodies and flow cytometric analysis". Cells were then fixed and stained for EdU using the Click-iT Plus EdU Alexa Fluor 647 Flow Cytometry Assay Kit (Thermo Fisher Scientific) following manufacturer's instructions. Flow cytometric analysis was then performed on an LSR Fortessa (BD).

**Statistics**. With the exception of RNAseq data, statistical analyses were performed with GraphPad Prism software. Statistical significance for normally distributed data was determined by two-tailed $t$ tests, adding the Welch's correction for groups with unequal variance, while data that were not normally distributed were analyzed with the non-parametric Mann–Whitney test. Statistical significance for RNAseq data was assessed using the Wald test with the Benjamini and Hochberg adjustment, and this is described in more details above under "RNAseq and gene pathway analyses". Data are represented as means ± SEM. Significance levels are labeled as *$P < 0.05$, **$P < 0.01$, ***$P < 0.001$, ****$P < 0.0001$; n.s. not significant ($P > 0.05$). All experimental replicates are biological, not technical. Investigators were not blinded to samples.

## Data availability

The RNAseq data have been deposited in NCBI's Gene Expression Omnibus[66] under the accession number GSE109147. All other data supporting the findings of these studies are available within the paper and the supplementary information files and/or from the corresponding author upon reasonable request.

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

## Acknowledgements

The authors acknowledge Pfizer as the source of the MRL/*lpr*-IL21R KO mice. We thank the NJH Genomics Core Service facility, and particularly Kendra Walton, for their technical help with the RNASeq studies. We acknowledge the NJH and University of Colorado Department of Immunology and Microbiology flow cytometry facilities for cell sorting and flow cytometry acquisition, and the Biological Resource Center at NJH and the Vivarium at the University of Colorado AMC for assistance with mouse husbandry. We are grateful to members of the Pelanda and Torres laboratories for their advice and valuable discussions and to Dr. Kathy Pape (University of Minnesota) for her advice on PE immunization and PE-reactive B cell staining. This study was supported by the National Institutes of Health grants AI052310 (to R.P.), AI052310-S1 (to R.P. and A.S.), AI052157 and AI078468 (to R. M.T.). It was also supported in part by a grant to R.P. from the Lupus Research Institute, Inc., and by the Cancer Center Support Grant P30CA046934 for shared resource. A.S. was also partly supported by the T32 AI074491 training grant.

## Author contributions

R.P. and A.S. designed the experiments and interpreted the data. R.M.T. contributed to data interpretation and experimental design. A.S. performed the experiments with technical help from J.N.P. B.P.O'.C. supervised the RNAseq data acquisition and S.M.L. and T.D. analyzed the RNAseq data. A.L.R. contributed the IL-21R KO MRL/*lpr* mouse. A.S. and R.P. wrote the manuscript with the editorial assistance of other authors.

## Additional information

**Competing interests:** The authors declare no competing interests.

