## [Peer Review File · Nature Communications]

Reviewers' comments:

Reviewer #1 (Remarks to the Author):

This manuscript is based on the knowledge reported previously by the authors (Journal of experimental medicine in 2012). They had shown the importance and the mechanism involved in the generation of dual- κ cells in a new engineered MRL/lpr mouse.

Using transcriptome approach, the authors further investigated which signaling paths are particularly important for the generation of dual-reactive B cells. They have focused on the IL-21, TLR7, and TLR9 pathways. For the TLR signaling pathway, they found that TLR9 dependent stimulation increases the expansion of dual-reactive B cells in vitro and in vivo. For the IL-21 signaling pathway, they further generated IL-21R deficient MRL/lpr-Igkm/h and showed that the number of those dual- κ B cells were reduced in the absence of IL-21 signaling.

Findings are novel and interesting, but they are descriptive and presented in a fragmented manner. Readers can understand the change in the frequency of dual- κ B cells in each compared experiment, but can not understand how relevant these pathways are in the expression of autoimmunity. The authors do not delve into revealing how each of the pathways promotes the expansion of dual- κ B cells.

Specific points:

1) The authors need to explain why they used B220 for gating B cells instead of CD19 in some experiments (Figure 1C-F, Figure 5, and Supplementary Figure 3A). B220 is strongly expressed on double negative T cells beside B cells in typical MRL/lpr mice. And those double negative T cells increase a lot (more than 60% of total T cells) and are considered to contribute disease pathogenesis. Thus using B220 only as a marker of B cells may affect the results of these experiments.

For example, in Figure 1, they need to clarify the gating strategy of single- κ and dual- κ B cells, and need to show that T cells are not included in these B220+ cells.

2) In Figure 5, the authors claim that the frequencies of both dual- κ B cells and dual- κ IgG class-switched B cells were decreased in IL-21R deficient MRL/lpr mice comparing to those in IL-21R sufficient counterparts. However, they did not show how these dual- κ B cell reductions could affect disease expression in MRL/lpr mice. Also the authors need to show evidence why these dual- κ B cells are reduced in the absence of the IL-21 pathway --reduction of clonal expansion in peripheral or decreased generation of dual- κ B cells in the bone marrow.

3) In Figure 6., the authors need to compare same observation using IL-21-R deficient MRL/lpr-Igkm/h mice to assess the contribution of IL-21 as written in result section.

4) In discussion, the authors claimed that increased frequency of dual-reactive B cells as a hallmark of disease but the statement is still speculative.

Reviewer #2 (Remarks to the Author):

This study explores some of the phenotypic and functional properties of dual light chain expressing B cells in an autoimmune prone mouse strain. The studies extend prior work by this and other groups that have associated these dual light chain expressing cells in autoimmune disease. The studies are generally well performed and clearly enunciated. For the most part the direct conclusions are valid. Nonetheless, the findings at this point are descriptive and none of the features revealed are explored in sufficient detail - either from a mechanistic standpoint or with regard to their relationship to the onset or progression of disease.

The overarching message from these descriptive analyses is that the dual kappa B cells have features generally associated with activated or antigen experienced B cells. The shortcoming of the current work is that the characterizations are performed on a single slice in time at which a lot of these cells have already acquired these characteristics. Thus, they do not yield insight as to when

and how they have done so, or why they are more likely than single-specificity B cells to do so. One or more of these questions would need to be pursued in some depth for the paper to achieve high priority. All would require substantial further experimental work.

For example, an important question is whether these are intrinsic features acquired during development and selection into preimmune pools by virtue of their dual specificity per se or via escape from selection, or are they acquired based on chronic or prior antigen-driven (self or otherwise) stimuli, such that dual specific cells simply have a higher likelihood of getting activated. In this regard, one wonders whether the assignment to "follicular" or "MZ" pools based on the staining criteria employed here may fail to resolve some residual heterogeneity. In fact, a lot of the phenotypic aspects reported, such as high MHC class II and CD80, would generally be used as criteria to classify a cell - regardless of CD21/24 phenotype - as 'activated' rather than follicular. Further criteria ought to be employed to assess this, and/or the existing features ought to be exploited to more extensively parse the dual kappa pools. For example, what is the IgM/IgD status of the dual-kappa cells that are being termed 'follicular' - are some of them IgD low to negative indicating prior antigen exposure? In the same vein, further separation and analysis based on these features within the dual kappa subsets might prove informative. For example, the flow plots show that the dual kappa cells are indeed shifted right for markers such as MHCII as the authors claim, but some of them are still in the range for the single kappa FO B cells. If one were to sort the high and low ends of these markers among the dual kappas, would the ones that have the more 'activated' (right shifted) phenotype be those driving the differences seen in the arrays and functional assays? If so, would the ones at the low end for these activation markers (overlapping with the single kappa 'folliculars') show similar response profiles (to CpG etc) to single kappas with the same phenotype? If this were the case, then these are not an intrinsic difference but reflect a disproportionate shuttling of dual kappas into the activated/antigen experience pools, rather than an intrinsic difference during development of selection into naïve preimmune pools.

2) A second question that needs to be addressed is which of these various features rely on the others how are they related - for example, is the IL21 requirement upstream of acquiring the activation markers and response characteristics, or vice-versa, or are these independent events. Sorting these out would likely require similarly detailed parsing of the dual kappas to establish such relationships.

We thank the Editors and the Reviewers for their interest in our study and for the comments that have led to enhance the quality of our manuscript.

Response to Reviewers' comments:

Reviewer #1:

1) The authors need to explain why they used B220 for gating B cells instead of CD19 in some experiments (Figure 1C-F, Figure 5, and Supplementary Figure 3A). B220 is strongly expressed on double negative T cells beside B cells in typical MRL/lpr mice. And those double negative T cells increase a lot (more than 60% of total T cells) and are considered to contribute disease pathogenesis. Thus using B220 only as a marker of B cells may affect the results of these experiments.

For example, in Figure 1, they need to clarify the gating strategy of single- κ and dual- κ B cells, and need to show that T cells are not included in these B220+ cells.

Response: we thank the reviewer for bringing up this issue. We were aware of the presence of B220+ T cells in MRL/lpr mice and in all our staining panels we have used different approaches to exclude T cells and/or exclusively gate on B cells. In some panels we used CD19 or CD22 to gate on B cells and not T cells. In others, we used a B220+CD3-gate to exclude T cells. And in some other panels where we used anti-B220 and could not add anti-CD3 antibodies, we used Igk to gate only on (Igk+) B cells. We have added a general sentence to state this in the Methods as well as revised the Figure legends.

2) In Figure 5, the authors claim that the frequencies of both dual- κ B cells and dual- κ IgG class-switched B cells were decreased in IL-21R deficient MRL/lpr mice comparing to those in IL-21R sufficient counterparts. However, they did not show how these dual- κ B cell reductions could affect disease expression in MRL/lpr mice. Also the authors need to show evidence why these dual- κ B cells are reduced in the absence of the IL-21 pathway -- reduction of clonal expansion in periphery or decreased generation of dual- κ B cells in the bone marrow.

Response: The disease progression of MRL/lpr IL21R KO mice has been well described by Andy Rankin (who is a coauthor here) in its publication in *The Journal of Immunology* in 2012. Whether the decrease of dual- κ cells is responsible for the disease outcome is a question of great importance to us, but also not a question easy to address. To address this, we have been studying mice with only one Igk allele, which are therefore unable to generate dual- κ cells. However, in these mice a compensatory mechanism leads to the expansion of kappa/lambda cells, preventing us to study lupus disease in the absence of dual-antibody B cells. Therefore, this is a question we are still working on and is outside the scope of this paper.

To address the question of why dual- κ cells are reduced in the absence of the IL-21 pathway, we have performed additional analyses. We have found that dual- κ cells are already reduced in the immature/transitional splenic B cell populations and, thus, before the mature cell stages (new Supplemental Fig. 3C). This reduction, which may be due to a previously described contribution of IL-21 to B cell development, likely explains the reduced dual- κ cell frequency in follicular and marginal zone B cells. Furthermore, we have treated IL-21R KO mice with EdU and found that in these mice while dual- κ cells still have higher proliferation rate than single- κ cells, this is much less than in wildtype mice (new Fig. 5F). Thus, the IL-21 pathway contributes to enhance the expansion of dual- κ cells over that of single- κ cells.

Interestingly, we did not find a contribution of IL-21 during the initial expansion of dual-k B cells to the T cell-dependent antigen PE (new Fig. 6D), a fact that actually makes sense given this expansion was in magnitude the same for single and dual-k cells in IL-21R sufficient mice. Thus, IL-21 may promote plasmablasts or memory dual-k cells to proliferate, but does not promote initial blasting of antigen-reactive B cells. We also cannot exclude an alternative explanation, an indirect effect on dual-k cell proliferation caused by a suppression of disease in IL-21R KO mice.

We should also mention that for reasons unclear to us our IL-21R KO MRL/lpr mouse colony breeds very poorly (this is actually a general problem we have with mouse breeding in our animal facility and that affects some strains more than others) and thus it is difficult to obtain a sufficient number of mice for experiments. Therefore, we were limited in the amount of analyses (and group size) we were able to do in response to the Reviewer comments on IL-21.

3) In Figure 6., the authors need to compare same observation using IL-21-R deficient MRL/lpr-Igkm/h mice to assess the contribution of IL-21 as written in result section.

Response: We have immunized a small group of IL-21R KO mice (see comment above about mouse availability issues) with PE in Alum as shown in Fig. 6. We have found that the expansion of dual-k B cells reactive to PE is similar in the presence or absence of IL-21R signaling. Although this finding may initially seem puzzling, it is actually in agreement with our previous data showing that single-k and dual-k PE-reactive cells expand to a similar extent with Alum. Thus, the unique characteristics dual-k cells display do not affect the initial primary expansion of B cells to a foreign protein antigen. It is the antigen-reactive clonal precursor frequency that causes the largest difference in single-k and dual-k primary expansion with Alum adjuvant. We have added results from experiments in which we tested the memory response to PE. Here we show that the clonal expansion of memory dual-k cells after recall immunization is significantly larger than that of single-k cells. Thus, we believe in a model in which a larger amount of dual-k cells than single-k cells are recruited into the memory and plasmablast/plasma cell fractions because of their enhanced cross-reactivity to antigens, including self-antigens. Given dual-k cells are more frequently reactive to nuclear antigens, this causes increased internalization of agonists for TLR7/9, whose signals amplify the response to self and foreign antigens. TLR signaling can also drive IFN α /b expression which activates further TLR expression and thus signaling, leading to a positive autocrine loop that has a larger effect on dual-k than single-k cells. Testing whether this model is correct is beyond the scope of this study.

4) In discussion, the authors claimed that increased frequency of dual-reactive B cells as a hallmark of disease but the statement is still speculative.

Response: we removed the claim.

Reviewer #2:

1) An important question is whether these are intrinsic features acquired during development and selection into preimmune pools by virtue of their dual specificity per se or via escape from selection, or are they acquired based on chronic or prior antigen-driven (self or otherwise) stimuli, such that dual specific cells simply have a higher likelihood of getting activated.

Response: This is a very important question and we thank the reviewer for asking it. To address this question we compared the activation state (CD80 and CD86 expression) of single and dual-k cells in immature and transitional B cells in bone marrow and spleen and found that dual-k cells already show a more activated state in immature/transitional cell fractions in bone marrow tissue (new Supplemental Fig. 1F). This argues that at least some of the unique intrinsic features of dual-k cells are already acquired during development, at least upon initial expression of a BCR. We and others have previously demonstrated that dual-k cells are more frequently autoreactive than single-k cells. Thus, these findings argue that in the autoimmune MRL/lpr (and MRL) background, immature B cells that react with self-antigens undergo activation and positive selection into the spleen more frequently than single-k cells because of their more frequent autoreactivity from the start.

1.a) In this regard, one wonders whether the assignment to "follicular" or "MZ" pools based on the staining criteria employed here may fail to resolve some residual heterogeneity. In fact, a lot of the phenotypic aspects reported, such as high MHC class II and CD80, would generally be used as criteria to classify a cell - regardless of CD21/24 phenotype - as 'activated' rather than follicular.

Response: We cannot exclude that the follicular and marginal zone B cell populations were not already enriched with activated cells in the dual-k cell samples we sorted for the RNAseq analyses, relative to single-k cell samples. But the point of this analysis was to determine what drives the existing dual-k population in MRL/lpr mice, irrespective of their activation or differentiation state (besides the follicular and marginal zone subdivision). To address the question of whether the dual-reactivity alters the gene expression profiling per se will require perform RNAseq analyses of single and dual-k cells in immature populations, or after the removal of activated or differentiated cells, and this is outside the scope of this study. However, we did perform some additional analyses that help understand this issue (see below in 1.b. and 1.c.).

1.b) Further criteria ought to be employed to assess this, and/or the existing features ought to be exploited to more extensively parse the dual kappa pools. For example, what is the IgM/IgD status of the dual-kappa cells that are being termed 'follicular' - are some of them IgD low to negative indicating prior antigen exposure?

Response: The dual-k "follicular" population has an IgM/IgD profile similar to that of single-k B cells (new figure panel in Supplemental Fig. 2C). When gated based on CD21, CD23, and CD24, both single and dual-k "follicular" populations have a 2-5% contamination of IgM-IgD- (switched or BCR-internalized?) cells. We compared the levels of IgM and IgD on the IgM+ follicular (CD23+ CD21+ CD24low) population: the dual-k cells have, on average, slightly lower IgD levels (about 10% less) and slightly higher IgM levels (about 20% more), though the difference is not statistically significant. These data have been added in Supplemental Fig. 2, C and D. In order to determine the potential "contamination" of memory B cells and plasmablasts in the cell samples used for the RNAseq analyses, we have analyzed the IgG and CD138 transcripts (Supplemental Fig. 1A). Based on this analysis, the single and dual-k cell samples were similarly contaminated by these cells. The samples obtained in the first sort appeared to contain a larger amount of effector cells in both the single and dual-k cell subsets, but the other two had less and were more similar. We also checked for the expression of many genes that have been described to be expressed (more or less) by memory B cells by the Shlomchik group (PMID: 24880458): we did not find significant differences between single and dual-k cell samples (not shown). Thus, overall we do not

believe that the results of the RNAseq analyses were significantly skewed by the way we sorted follicular and marginal zone B cells.

1.c) In the same vein, further separation and analysis based on these features within the dual kappa subsets might prove informative. For example, the flow plots show that the dual kappa cells are indeed shifted right for markers such as MHCII as the authors claim, but some of them are still in the range for the single kappa FO B cells. If one were to sort the high and low ends of these markers among the dual kappas, would the ones that have the more 'activated' (right shifted) phenotype be those driving the differences seen in the arrays and functional assays? If so, would the ones at the low end for these activation markers (overlapping with the single kappa 'folliculars') show similar response profiles (to CpG etc) to single kappas with the same phenotype? If this were the case, then these are not an intrinsic difference but reflect a disproportionate shuttling of dual kappas into the activated/antigen experience pools, rather than an intrinsic difference during development of selection into naïve preimmune pools.

Response: we thank the reviewer for this suggestion, which we followed. We have magnetically purified naïve splenic B cells by depleting CD43+CD80+CD86+ cells confirming the negative cells were depleted of CD80+CD86+ activated cells, as shown in new Supplemental Fig. 1G (this is also true for MHC class II, not shown). We then measured the response of single and dual-k naïve B cells to CpG, R848, and IFN- α . We found that dual-k naïve cells still show heightened response to these stimuli (new Fig. 3, F and G). Thus, the enhanced activation or expansion dual-k cells display in response to innate stimuli is intrinsic and acquired at their origin, independent of their activation state.

2) A second question that needs to be addressed is which of these various features rely on the others, how are they related - for example, is the IL21 requirement upstream of acquiring the activation markers and response characteristics, or vice-versa, or are these independent events. Sorting these out would likely require similarly detailed parsing of the dual kappas to establish such relationships.

Response: as detailed in the response to Reviewer 1 above, we have performed some additional analyses of IL-21R KO mice (which unfortunately breed poorly). We have found that the IL-21 pathway is not required for the enhanced activation to dual-k cells in immature or transitional B cell population (new Supplemental Fig. 3C). It does however play an important role in the expansion of dual-k B cells in all cell subsets including the immature/transitional cell subset (new Fig. 5C). Based on these findings we propose that IL-21 contributes to the expansion of dual-k cells in transitional-mature cell population by either affecting immature B cell development or, in an indirect way, by promoting disease and causing inflammation.

**In addition, in revising the manuscript we noticed a couple of errors that we have now revised: in Fig. 3 D, E we had inadvertently switched the data for IFN- α and IFN- γ stimulation. These are now been properly reported in the revised Figure. The graph in Fig. 4G listed on the y-axis the MFI of CD21 instead of IL-21R. We have corrected this typo. We have added an analysis of memory B cell response to PE (Fig. 6, G,H) that shows superior response of dual-k over single-k B cells. Finally, we have revised the title to reflect a more general contribution of T cell-dependent and independent signals in promoting dual-k B cells.

REVIEWERS' COMMENTS:

Reviewer #1 (Remarks to the Author):

Thank you for addressing carefully all comments.

Reviewer #2 (Remarks to the Author):

The authors have made a good-faith attempt to respond to the prior critiques, and have added further analyses to address some of the more substantive concerns. These additional experiments have clarified some of the prior findings and led to amendment of the major message of the paper. Overall, the role of IL21 signaling in the favoring of dual kappa B cell activity is diminished by the added data from the knockout mice. This clarification is worthwhile, but reduces the mechanistic and conceptual impact of the paper, since it looks as if the basis for the activated phenotype of the dual kappa expressing B cells remains unclear. Nonetheless, a thorough description of these cells and the factors likely driving them in vivo is worthwhile.

An important observation is that this partially activated phenotype seems to occur as early as the immature BM/transitional SPL stages of development. This is an important conceptual point if it has been interpreted correctly, since it indeed suggests either a higher likelihood of escape from deletion or enhanced positive selection, as the authors suggest. The data supporting this are largely based on surface phenotype - so it remains possible that lowered CD23, IgD, and CD21 are activation-driven events that lead to these cells looking like immature and transitional but in reality reflect activation within mature pools (CD24 can also be upregulated on activation). The authors might want to note this as an alternative possibility in the text, or perform experiments that would delineate these possibilities.

The authors might want to take a look at their dual kappa versus single kappa B cells for CD11c and/or Tbet expression, since B cells with these features have very similar phenotype to those described for the dual kappas here, are also driven preferentially by nucleic acid sensing TLRs and IL21/IFN γ , and are associated with humoral autoimmunity in mouse models and human disease (Rubtsova et al and Hao et al, Blood, 2011; Naradikian et al J Immunol 2016, Wang et al Nature Comm 2018, Manni et al Nat Immunol), . If enriched for this phenotype, it would suggest the dual kappas have a higher propensity to adopt this fate.

We thank the Editors and the Reviewers for their interest in our study and for the comments that have led to enhance the quality of our manuscript.

Response to Reviewers' comments:

Reviewer #1:

1) The authors need to explain why they used B220 for gating B cells instead of CD19 in some experiments (Figure 1C-F, Figure 5, and Supplementary Figure 3A). B220 is strongly expressed on double negative T cells beside B cells in typical MRL/lpr mice. And those double negative T cells increase a lot (more than 60% of total T cells) and are considered to contribute disease pathogenesis. Thus using B220 only as a marker of B cells may affect the results of these experiments.

For example, in Figure 1, they need to clarify the gating strategy of single- κ and dual- κ B cells, and need to show that T cells are not included in these B220+ cells.

Response: we thank the reviewer for bringing up this issue. We were aware of the presence of B220+ T cells in MRL/lpr mice and in all our staining panels we have used different approaches to exclude T cells and/or exclusively gate on B cells. In some panels we used CD19 or CD22 to gate on B cells and not T cells. In others, we used a B220+CD3-gate to exclude T cells. And in some other panels where we used anti-B220 and could not add anti-CD3 antibodies, we used Igk to gate only on (Igk+) B cells. We have added a general sentence to state this in the Methods as well as revised the Figure legends.

2) In Figure 5, the authors claim that the frequencies of both dual- κ B cells and dual- κ IgG class-switched B cells were decreased in IL-21R deficient MRL/lpr mice comparing to those in IL-21R sufficient counterparts. However, they did not show how these dual- κ B cell reductions could affect disease expression in MRL/lpr mice. Also the authors need to show evidence why these dual- κ B cells are reduced in the absence of the IL-21 pathway -- reduction of clonal expansion in periphery or decreased generation of dual- κ B cells in the bone marrow.

Response: The disease progression of MRL/lpr IL21R KO mice has been well described by Andy Rankin (who is a coauthor here) in its publication in *The Journal of Immunology* in 2012. Whether the decrease of dual- κ cells is responsible for the disease outcome is a question of great importance to us, but also not a question easy to address.

To address the question of why dual- κ cells are reduced in the absence of the IL-21 pathway, we have performed additional analyses. We have found that dual- κ cells are already reduced in the immature/transitional splenic B cell populations and, thus, before the mature cell stages (new Supplemental Fig. 3C). This reduction, which may be due to a previously described contribution of IL-21 to B cell development, likely explains the reduced dual- κ cell frequency in follicular and marginal zone B cells. Furthermore, we have treated IL-21R KO mice with EdU and found that in these mice while dual- κ cells still have higher proliferation rate than single- κ cells, this is much less than in wildtype mice (new Fig. 5F). Thus, the IL-21 pathway contributes to enhance the expansion of dual- κ cells over that of single- κ cells.

Interestingly, we did not find a contribution of IL-21 during the initial expansion of dual-k B cells to the T cell-dependent antigen PE (new Fig. 6D), a fact that actually makes sense given this expansion was in magnitude the same for single and dual-k cells in IL-21R sufficient mice. Thus, IL-21 may promote plasmablasts or memory dual-k cells to proliferate, but does not promote initial blasting of antigen-reactive B cells. We also cannot exclude an alternative explanation, an indirect effect on dual-k cell proliferation caused by a suppression of disease in IL-21R KO mice.

We should also mention that for reasons unclear to us our IL-21R KO MRL/lpr mouse colony breeds very poorly [REDACTED]

[REDACTED] and thus it is difficult to obtain a sufficient number of mice for experiments. Therefore, we were limited in the amount of analyses (and group size) we were able to do in response to the Reviewer comments on IL-21.

3) In Figure 6., the authors need to compare same observation using IL-21-R deficient MRL/lpr-Igkm/h mice to assess the contribution of IL-21 as written in result section.

Response: We have immunized a small group of IL-21R KO mice (see comment above about mouse availability issues) with PE in Alum as shown in Fig. 6. We have found that the expansion of dual-k B cells reactive to PE is similar in the presence or absence of IL-21R signaling. Although this finding may initially seem puzzling, it is actually in agreement with our previous data showing that single-k and dual-k PE-reactive cells expand to a similar extent with Alum. Thus, the unique characteristics dual-k cells display do not affect the initial primary expansion of B cells to a foreign protein antigen. It is the antigen-reactive clonal precursor frequency that causes the largest difference in single-k and dual-k primary expansion with Alum adjuvant. We have added results from experiments in which we tested the memory response to PE. Here we show that the clonal expansion of memory dual-k cells after recall immunization is significantly larger than that of single-k cells. Thus, we believe in a model in which a larger amount of dual-k cells than single-k cells are recruited into the memory and plasmablast/plasma cell fractions because of their enhanced cross-reactivity to antigens, including self-antigens. Given dual-k cells are more frequently reactive to nuclear antigens, this causes increased internalization of agonists for TLR7/9, whose signals amplify the response to self and foreign antigens. TLR signaling can also drive IFN α /b expression which activates further TLR expression and thus signaling, leading to a positive autocrine loop that has a larger effect on dual-k than single-k cells. Testing whether this model is correct is beyond the scope of this study.

4) In discussion, the authors claimed that increased frequency of dual-reactive B cells as a hallmark of disease but the statement is still speculative.

Response: we removed the claim.

Reviewer #2:

1) An important question is whether these are intrinsic features acquired during development and selection into preimmune pools by virtue of their dual specificity per se or via escape from selection, or are they acquired based on chronic or prior antigen-driven (self or otherwise) stimuli, such that dual specific cells simply have a higher likelihood of getting activated.

Response: This is a very important question and we thank the reviewer for asking it. To address this question we compared the activation state (CD80 and CD86 expression) of single and dual-k cells in immature and transitional B cells in bone marrow and spleen and found that dual-k cells already show a more activated state in immature/transitional cell fractions in bone marrow tissue (new Supplemental Fig. 1F). This argues that at least some of the unique intrinsic features of dual-k cells are already acquired during development, at least upon initial expression of a BCR. We and others have previously demonstrated that dual-k cells are more frequently autoreactive than single-k cells. Thus, these findings argue that in the autoimmune MRL/lpr (and MRL) background, immature B cells that react with self-antigens undergo activation and positive selection into the spleen more frequently than single-k cells because of their more frequent autoreactivity from the start.

1.a) In this regard, one wonders whether the assignment to "follicular" or "MZ" pools based on the staining criteria employed here may fail to resolve some residual heterogeneity. In fact, a lot of the phenotypic aspects reported, such as high MHC class II and CD80, would generally be used as criteria to classify a cell - regardless of CD21/24 phenotype - as 'activated' rather than follicular.

Response: We cannot exclude that the follicular and marginal zone B cell populations were not already enriched with activated cells in the dual-k cell samples we sorted for the RNAseq analyses, relative to single-k cell samples. But the point of this analysis was to determine what drives the existing dual-k population in MRL/lpr mice, irrespective of their activation or differentiation state (besides the follicular and marginal zone subdivision). To address the question of whether the dual-reactivity alters the gene expression profiling per se will require perform RNAseq analyses of single and dual-k cells in immature populations, or after the removal of activated or differentiated cells, and this is outside the scope of this study. However, we did perform some additional analyses that help understand this issue (see below in 1.b. and 1.c.).

1.b) Further criteria ought to be employed to assess this, and/or the existing features ought to be exploited to more extensively parse the dual kappa pools. For example, what is the IgM/IgD status of the dual-kappa cells that are being termed 'follicular' - are some of them IgD low to negative indicating prior antigen exposure?

Response: The dual-k "follicular" population has an IgM/IgD profile similar to that of single-k B cells (new figure panel in Supplemental Fig. 2C). When gated based on CD21, CD23, and CD24, both single and dual-k "follicular" populations have a 2-5% contamination of IgM-IgD- (switched or BCR-internalized?) cells. We compared the levels of IgM and IgD on the IgM+ follicular (CD23+ CD21+ CD24low) population: the dual-k cells have, on average, slightly lower IgD levels (about 10% less) and slightly higher IgM levels (about 20% more), though the difference is not statistically significant. These data have been added in Supplemental Fig. 2, C and D. In order to determine the potential "contamination" of memory B cells and plasmablasts in the cell samples used for the RNAseq analyses, we have analyzed the IgG and CD138 transcripts (Supplemental Fig. 1A). Based on this analysis, the single and dual-k cell samples were similarly contaminated by these cells. The samples obtained in the first sort appeared to contain a larger amount of effector cells in both the single and dual-k cell subsets, but the other two had less and were more similar. We also checked for the expression of many genes that have been described to be expressed (more or less) by memory B cells by the Shlomchik group (PMID: 24880458): we did not find significant differences between single and dual-k cell samples (not shown). Thus, overall we do not

believe that the results of the RNAseq analyses were significantly skewed by the way we sorted follicular and marginal zone B cells.

1.c) In the same vein, further separation and analysis based on these features within the dual kappa subsets might prove informative. For example, the flow plots show that the dual kappa cells are indeed shifted right for markers such as MHCII as the authors claim, but some of them are still in the range for the single kappa FO B cells. If one were to sort the high and low ends of these markers among the dual kappas, would the ones that have the more 'activated' (right shifted) phenotype be those driving the differences seen in the arrays and functional assays? If so, would the ones at the low end for these activation markers (overlapping with the single kappa 'folliculars') show similar response profiles (to CpG etc) to single kappas with the same phenotype? If this were the case, then these are not an intrinsic difference but reflect a disproportionate shuttling of dual kappas into the activated/antigen experience pools, rather than an intrinsic difference during development of selection into naïve preimmune pools.

Response: we thank the reviewer for this suggestion, which we followed. We have magnetically purified naïve splenic B cells by depleting CD43+CD80+CD86+ cells confirming the negative cells were depleted of CD80+CD86+ activated cells, as shown in new Supplemental Fig. 1G (this is also true for MHC class II, not shown). We then measured the response of single and dual-k naïve B cells to CpG, R848, and IFN- α . We found that dual-k naïve cells still show heightened response to these stimuli (new Fig. 3, F and G). Thus, the enhanced activation or expansion dual-k cells display in response to innate stimuli is intrinsic and acquired at their origin, independent of their activation state.

2) A second question that needs to be addressed is which of these various features rely on the others, how are they related - for example, is the IL21 requirement upstream of acquiring the activation markers and response characteristics, or vice-versa, or are these independent events. Sorting these out would likely require similarly detailed parsing of the dual kappas to establish such relationships.

Response: as detailed in the response to Reviewer 1 above, we have performed some additional analyses of IL-21R KO mice (which unfortunately breed poorly). We have found that the IL-21 pathway is not required for the enhanced activation to dual-k cells in immature or transitional B cell population (new Supplemental Fig. 3C). It does however play an important role in the expansion of dual-k B cells in all cell subsets including the immature/transitional cell subset (new Fig. 5C). Based on these findings we propose that IL-21 contributes to the expansion of dual-k cells in transitional-mature cell population by either affecting immature B cell development or, in an indirect way, by promoting disease and causing inflammation.

**In addition, in revising the manuscript we noticed a couple of errors that we have now revised: in Fig. 3 D, E we had inadvertently switched the data for IFN- α and IFN- γ stimulation. These are now been properly reported in the revised Figure. The graph in Fig. 4G listed on the y-axis the MFI of CD21 instead of IL-21R. We have corrected this typo. We have added an analysis of memory B cell response to PE (Fig. 6, G,H) that shows superior response of dual-k over single-k B cells. Finally, we have revised the title to reflect a more general contribution of T cell-dependent and independent signals in promoting dual-k B cells.

Second round comments:

Reviewer 2 raised the point that the B cell population we defined as immature/transitional (based on lowered CD23, IgD, and CD21, and increased CD24) might contain activated mature B cells that account for the observations we describe. Though this remains a possibility, we argue that only a small fraction of activated B cells would show CD23 and IgD downregulation plus CD24 upregulation as the population we gate in the bone marrow, or CD21 downregulation and CD24 upregulation as the population we gate in the spleen. Thus, these cells would be in very small numbers relative to the main immature/transitional cell subsets we gated, and that also appear to be in proportion expected in bone marrow and spleen. These potential contaminating active mature B cells would be in too small numbers to account for the differences we observed in CD80/CD86 expression.

Reviewer 2 also raised the point that dual-k B cells resemble CD11c/Tbet-positive age associated B cells. While we have not analyzed the expression of CD11c and Tbet on dual-k B cells, many years ago we measured the frequency of dual-k B cells in the ABC B cell population identified by the expression of intermediate levels of CD11b. These analyses indicate that the frequency of dual-k cells in the ABC subset is similar to that of MZB (about 10%) and, based on the frequency of ABCs in the B cell population, we estimate that only a minor fraction of dual-k B cells overlap with ABCs.

Response to additional comments from Reviewer 2:

Reviewer 2 raised the point that the B cell population we defined as immature/transitional (based on lowered CD23, IgD, and CD21, and increased CD24) might contain activated mature B cells that account for the observations we describe.

Response: Though this remains a possibility, we argue that only a small fraction of activated B cells would show CD23 and IgD downregulation plus CD24 upregulation as the population we gate in the bone marrow, or CD21 downregulation and CD24 upregulation as the population we gate in the spleen. Thus, these cells would be in very small numbers relative to the main immature/transitional cell subsets we gated, and that also appear to be in proportion expected in bone marrow and spleen. These potential contaminating active mature B cells would be in too small numbers to account for the differences we observed in CD80/CD86 expression.

Reviewer 2 also raised the point that dual-k B cells resemble CD11c/Tbet-positive age associated B cells.

Response: While we have not analyzed the expression of CD11c and Tbet on dual-k B cells, many years ago we measured the frequency of dual-k B cells in the ABC B cell population identified by the expression of intermediate levels of CD11b. These analyses indicate that the frequency of dual-k cells in the ABC subset is similar to that of MZB (about 10%) and, based on the frequency of ABCs in the B cell population, we estimate that only a minor fraction of dual-k B cells overlap with ABCs.